# TGV-KV: Text-Grounded KV Eviction for Vision-Language Models

**Jizhihui Liu** [1]   **Ruizi Han** [1]   **Miao Zhang** [1 2]   **Rui Shao** [1]   **Xuebo Liu** [1]   **Weili Guan** [1 2]   **Yaowei Wang** [1 2]

## Abstract

Vision-Language Models (VLMs) inherit the auto-regressive generation paradigm and cache the keys and values (KV) of all previous tokens to accelerate inference, resulting in memory consumption that scales linearly with context length. This issue is particularly pronounced in VLMs due to substantial redundancy in the visual modality. Although KV cache eviction approaches can effectively reduce inference memory, they often incur significant performance degradation in VLMs, as most are designed for language models and overlook the inherent gap between text and vision. By systematically analyzing the modality gap in VLMs in this work, we argue that the importance of visual information should be grounded in textual guidance and accordingly propose a **T**ext-**G**rounded KV Eviction method for **V**LMs (**TGV-KV**). TGV-KV comprises three submodules: *(1) Text-Vision Budgeting (TVB)* assigns budget to each layer based on the mutual information interaction. *(2) Text-Weighted Ranking (TWR)* assesses the priority of text and ranks vision importance based on weighted text-image attention. *(3) Text-Prioritised Retention (TPR)* policy strategically preserves text KV to avoid acute information loss. We evaluate TGV-KV across five models with different sizes and architectures, showing that TGV-KV preserves 99.2% full-KV accuracy on the VizWiz-VQA task with LLaVA-NeXT and boosts end-to-end throughput by 52.6% with an extreme retention budget of 5%. Code Link.

## 1. Introduction

Vision-Language Models (VLMs) (Bai et al., 2025; Liu et al., 2024; 2023) have revolutionized multimodal understanding and visual reasoning in recent years. Inheriting

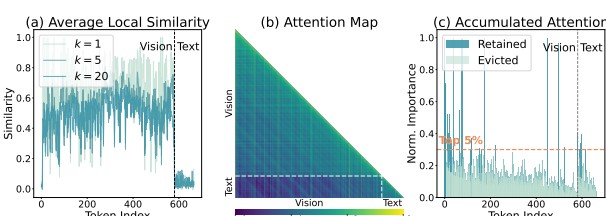

*Figure 1.* **Visualization and Consequence of Modality Gap.** **(a)** The average cosine similarity of vision and text tokens with $k$ neighbours. **(b)** The layer 2 attention map (in log scale) in LLaVA, the intra-modality part (text-text, vision-vision) is much more intense than inter-modality (text-vision). **(c)** Accumulated attention score across all the layers, the eviction is highly uneven and most text KV pairs are evicted by cumulative attention score.

the auto-regressive architecture of Large Language Models (LLMs) (Grattafiori et al., 2024; Yang et al., 2025a), VLMs rely on the Key-Value (KV) cache mechanism to eliminate redundant computations and accelerate generation. However, the KV cache grows linearly with context length, incurring severe bottlenecks in memory consumption and inference latency. This challenge is more exacerbated in the multimodal setting since high-resolution images and long videos often take up thousands of tokens (Lin et al., 2024). Despite the large quantity, many studies (Chen et al., 2024a; Liu et al., 2025a) have found that most of these vision features are highly redundant and removing a large proportion of them negligibly degrades the model performance.

To overcome the memory issue in VLM, most current methods (Chen et al., 2024a; Liu et al., 2025a; Yang et al., 2025b) prune redundant vision tokens directly before or during the prefill at a time. However, these methods often influence model performance sharply since once a token is pruned, it is no longer accessed in subsequent model layers or decoding steps, and the contained unextracted information is permanently lost. With higher flexibility, KV cache eviction serves as a promising mitigation. However, such methods face critical performance degradation in VLM since most are designed for language models. To explain this, we identify three significant differences between text and vision and attribute this drop to the modality gap. **(1)** Vision tokens are very similar to each other, while text tokens are very diverse (Fig. 1 (a)). **(2)** This inherent gap causes a low-attention area in the text-vision part, making the unimodal attention

---

[1]Harbin Institute of Technology, Shenzhen [2]Peng Cheng Laboratory. Correspondence to: Miao Zhang <zhangmiao@hit.edu.cn>.

*Proceedings of the 43rd International Conference on Machine Learning*, Seoul, South Korea. PMLR 306, 2026. Copyright 2026 by the author(s).

fluctuate (Fig. 1 (b)). **(3)** This special distribution causes a sharp shift in accumulated attention at the intersection of vision and text, making the eviction extremely uneven (Fig. 1 (c)). Therefore, directly transferring current KV cache eviction methods from LLM to VLM overlooks the spatial redundancy of vision and the mutual modality interaction, suffering from the inconsistent attention distribution.

In this paper, we take a systematic study of the multimodal attention pattern in VLM and aim to elucidate the role of each component in KV cache eviction. Specifically, we reveal three key observations through extensive experiments, including text-vision attention's effectiveness in layer budget division, an appropriate indicator to measure multimodal KV importance, and the relative priority of different modalities during eviction. Based on these key findings, we propose a robust and **T**ext-**G**rounded KV eviction framework for **V**LMs, i.e., TGV[1]-KV. The key idea of TGV-KV is to overcome the modality inconsistency issue by taking full advantage of text features. To this end, we propose three synergistic components: a layer budget allocation policy **Text-Vision Budgeting (TVB)**, a multimodal KV importance judge **Text-Weighted Ranking (TWR)**, and an eviction criterion **Text-Prioritised Retention (TPR)**. After the prefill, TVB extracts the text-vision attention and normalizes the summation layer-wisely to divide the total budget. TWR first evaluates the significance of each text token and applies a positional average to obtain the weight coefficient for vision token importance judgment. TPR adaptively evicts multimodal KV pairs based on the importance score while always keeping text KVs as long as the budget allows. These three modules take full consideration of text modality and maintain cross-modality consistency sufficiently, achieving outstanding performance in accuracy and efficiency.

We evaluate TGV-KV across diverse popular VLMs, covering basic LLaVA-series (Liu et al., 2023; 2024) to state-of-the-art Qwen-series (Bai et al., 2025). With an extreme compression ratio of **5%**, TGV-KV retains **92.5%** full-KV accuracy in DocVQA on Qwen3-VL-8B, while surpassing the best baseline by **33.0%** on LLaVA-NeXT, along with a **95%** reduction in memory and **52.6%** acceleration.

Our main contributions are summarized as follows:

- We take a systematic study of the attention pattern in VLMs and propose TGV-KV, an out-of-the-box KV eviction approach for multimodal KV eviction.

- We design three modules or policies, i.e., TVB, TWR, and TPR, to allocate budget, rank multimodal importance, and evict KV pairs while preserving performance, which can be adopted by subsequent VLM KV eviction studies seamlessly.

---

[1]TGV is also the abbreviation of Train à Grande Vitesse, the France high-speed railway system.

- Extensive studies across multiple VLMs demonstrate our TGV-KV achieves outstanding performance and substantial memory reduction, outperforming existing methods in both accuracy and efficiency.

## 2. Related Works

### 2.1. Vision-Language Models

Vision-Language Models (Liu et al., 2023; Achiam et al., 2023; Team et al., 2023) are usually composed of a Large Language Model decoder (Grattafiori et al., 2024; Yang et al., 2025a), a vision encoder (Radford et al., 2021; Zhai et al., 2023) and an adaptor (Alayrac et al., 2022). The vision encoder is a Vision Transformer (Dosovitskiy et al., 2021) that encodes visual inputs into abundant visual tokens, which are then projected into the text semantic space by the adaptor. The projected vision features are then concatenated with text embeddings, forming a unified multimodal input sequence. In the LLM decoder, all the counterparts are treated in the same way, performing unified causal self-attention (Vaswani et al., 2017) in each layer. During inference, VLM inherits the auto-regressive generation manner of LLM and stores the KVs of past tokens to accelerate generation. The memory consumption of KV cache grows linearly with the input sequence. For VLMs that adopt a dynamic-resolution vision encoder (Bai et al., 2025; Chen et al., 2024b), a video may take tens of thousands of tokens, incurring severe KV cache memory consumption.

### 2.2. Efficient Inference for VLMs

Many approaches have been proposed to overcome the memory burden in VLM generation, which can be roughly categorized into token pruning and KV eviction. Token pruning is usually applied before or during the prefill, which prunes tokens at a time. FastV (Chen et al., 2024a) and VisionZip (Yang et al., 2025b) prune tokens based on the attention score, while CDPruner (Zhang et al., 2025a) utilizes the text embeddings as a guidance. These methods lack flexibility and suffer from severe information loss. KV cache eviction evicts KV after the prefill and compresses the memory consumption during the decode phase. $H_2O$ (Zhang et al., 2023), SnapKV (Li et al., 2024b) assess KV importance by attention score, PyramidKV (Cai et al., 2024), SparseMM (Wang et al., 2025b), Ada-KV (Feng et al., 2024) allocate dynamic budget to layers or heads. Most of these methods do not consider the modality gap and handle different modalities as the same, which is suboptimal. AirCache (Huang et al., 2025) identifies key text tokens and judges vision importance with them, however, this requires extra computation and lacks mutual information flow analyses during budget allocation. In this work, we solve the mentioned problems by introducing text ground and mutual information flow during both budget allocation and importance assessment.

# 3. Method

In this section, we first revisit the principle of KV cache, and then present three key observations. Finally, we show the design of TGV-KV and depict it in Fig. 3.

## 3.1. Preliminary

Similar to LLM, the generation of a VLM can be divided into two phases, i.e., *Prefill* and *Decode*. Eviction methods are applied before the prefill phase and during the decode phase to control the total budget of KV cache.

**Prefill Phase.** The vision encoder and text tokenizer first encode multimodal inputs and text prompts into visual tokens $\mathbf{X_v} \in \mathbb{R}^{N_v \times d}$ and text tokens $\mathbf{X_t} \in \mathbb{R}^{N_t \times d}$, where $N_v$ and $N_t$ denote the number of vision and text tokens, $d$ denotes the model dimension. All the counterparts are then concatenated into a unified sequence $\mathbf{X} \in \mathbb{R}^{(N_v+N_t) \times d}$. In the self-attention of each layer, the model calculates

$$\mathbf{Q} = \mathbf{XW_Q}, \mathbf{K} = \mathbf{XW_K}, \mathbf{V} = \mathbf{XW_V}, \quad (1)$$

where $\mathbf{W_Q}, \mathbf{W_K}, \mathbf{W_V} \in \mathbb{R}^{d \times d}$ are pretrained projection matrices. The self-attention is then calculated by

$$\mathbf{A} = \mathrm{softmax}(\mathbf{QK^\top} + \mathbf{M}) \in \mathbb{R}^{(N_v+N_t) \times (N_v+N_t)}, \quad (2)$$

where $\mathbf{M}$ is an upper triangular matrix filled with $-\infty$ to avoid information leak. Since the key and value of context tokens are accessed at each decoding step, all the $\mathbf{K}$ and $\mathbf{V}$ of each layer are cached to avoid re-computation.

**Decode Phase.** During per-token generation, the input of time step $t$ is a single token $\mathbf{x} \in \mathbb{R}^d$. The key and value of $\mathbf{x}$ are stored in the cache by

$$\mathbf{K} = \mathrm{concat}(\mathbf{K}, \mathbf{xW_k}) \in \mathbb{R}^{(N_v+N_t+t) \times d}, \quad (3)$$

$$\mathbf{V} = \mathrm{concat}(\mathbf{V}, \mathbf{xW_v}) \in \mathbb{R}^{(N_v+N_t+t) \times d}. \quad (4)$$

The memory footprint of $\mathbf{K}$ and $\mathbf{V}$ grows linearly with $t$. As the context grows longer, this issue can not be ignored.

## 3.2. Key Insights

KV cache eviction methods commonly include two procedures: *budget allocation* and *KV eviction*. In this subsection, we conduct experiments to identify different attention pattern combinations in layer budgeting and eviction criteria.

**Inter-Layer Budget Allocation.** Intuitively, different layers handle different semantics, thus need different budgets. Unlike previous works with a fixed rule-based layer budget allocation policy (Chen et al., 2024a; Cai et al., 2024), we aim to assign budgets dynamically based on the attention score since tokens interact with each other in self-attention.

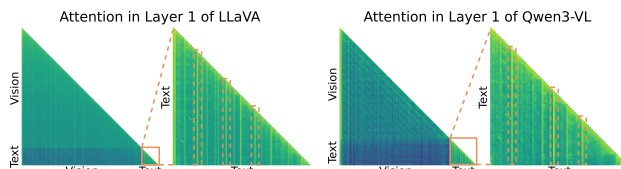

*Figure 2.* **Attention Map of VLMs.** We average all the heads and plot the attention map in LLaVA-1.5-7B and Qwen3-VL-8B. Each zoomed-in area denotes the text-text attention in a multimodal input. We label a few dominant text token in **orange box**. More visualizations can be found in **Appendix** C.1.

*Table 1.* **Eviction Performance with Different Policies.** We evaluate different layer budget allocation policies and KV eviction criteria on LLaVA. Note that even if evicting text first in (c), the system tokens are always kept. Percentage denotes retention ratio.

| Attention Pattern | ChartQA↑ | | | TextVQA$^{\text{lite}}$ ↑ | | |
|---|---|---|---|---|---|---|
| | 50% | 20% | 5% | 50% | 20% | 5% |
| Vanilla Model | 18.0 | 18.0 | 18.0 | 47.9 | 47.9 | 47.9 |
| *(a) Same Eviction Criterion (TV+TT) + Different Layer Budget* | | | | | | |
| Uniform | 17.8 | 17.8 | 13.9 | 47.9 | 47.7 | 32.8 |
| Vision-Vision (VV) | 17.7 | 17.6 | 14.1 | 48.3 | 47.4 | 34.2 |
| Text-Text (TT) | 17.8 | 17.9 | 14.2 | 48.3 | 46.9 | 36.1 |
| Text-Vision (TV) | 17.8 | 17.8 | 14.3 | 48.3 | 47.5 | 36.4 |
| *(b) Same Layer Budget (Uniform) + Different Importance Criteria* | | | | | | |
| Observation Window | 17.9 | 16.6 | 0.4 | 46.3 | 36.4 | 8.7 |
| Self-Attention | 4.7 | 3.9 | 4.8 | 31.8 | 28.4 | 23.5 |
| VV+TT Attention | 4.8 | 3.7 | 4.6 | 32.3 | 28.4 | 22.8 |
| TV+TT Attention | 17.9 | 17.7 | 11.0 | 48.4 | 47.8 | 37.3 |
| *(c) Same Layer Budget (Uniform) + Different Eviction Modalities* | | | | | | |
| Evict Vision First | 16.7 | 14.8 | 10.0 | 46.3 | 40.4 | 31.0 |
| Evict Text First | 0.2 | 0.2 | 0.2 | 4.4 | 4.4 | 4.4 |

Note that we currently do not consider head-wise analyses since unpacking heads breaks the parallelism of attention computation and affects efficiency.

We mainly consider three types of attention as guidance, i.e., vision-vision (VV), text-vision (TV), and text-text (TT). Specifically, given the set $\mathcal{A}^{(\mathrm{x})} = \{\mathbf{A}_l^{(\mathrm{x})}\}_{l=1}^L$ for each mode $\mathrm{x} \in \{\mathrm{VV}, \mathrm{TV}, \mathrm{TT}\}$, where $L$ denotes the number of layers in the decoder, we define the budget $b_l^{(\mathrm{x})}$ for each layer as:

$$b_l^{(\mathrm{x})} = \frac{\sum_{i,j}[\mathbf{A}_l^{(\mathrm{x})}]_{ij}}{\sum_{l'=1}^L \sum_{i,j}[\mathbf{A}_{l'}^{(\mathrm{x})}]_{ij}}. \quad (5)$$

As shown in **block (a)** of Table 1, Observation 3.1. We hypothesize that the TV attention implies the intensity of information exchange, serving as an ideal indicator of budget allocation since higher exchange needs more KV retention.

**Observation 3.1.** The intensity of cross-modal attention serves as a proxy for semantic fusion, positively correlated with a layer's demand for KV retention budget.

**Intra-Layer KV Eviction.** The remaining main problem is to evict "low-importance" KV pairs. We evaluate using an

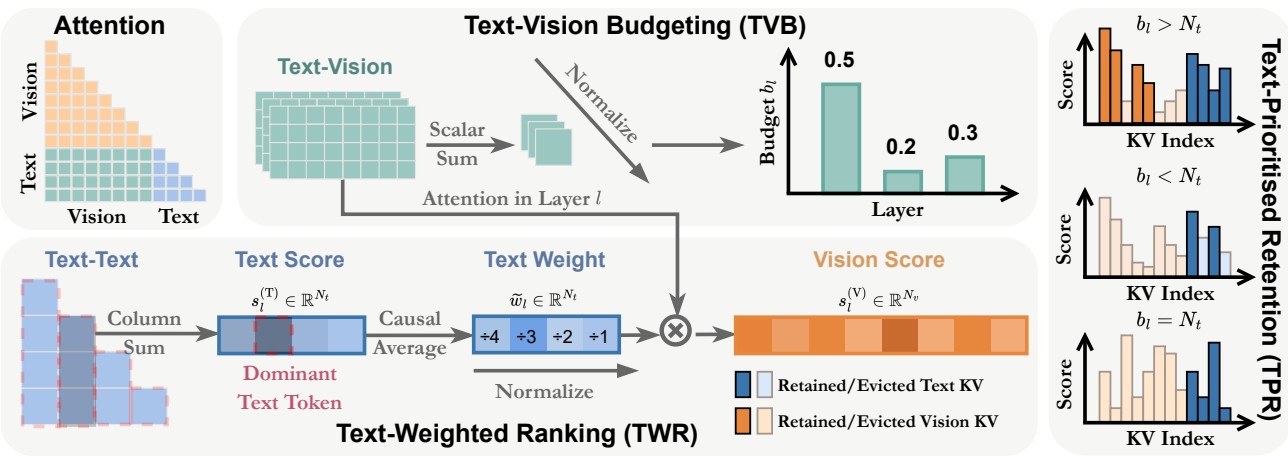

Figure 3. **Overview of TGV-KV framework.** Our method consists of three submodules or policies: **(1) Text-Vision Budgeting** adaptively allocates layer-wise KV budgets via text-image attention summation; **(2) Text-Weighted Ranking** determines vision importance score by re-weighting text-image attention with averaged text attention, and text importance by attention summation; **(3) Text-Prioritised Retention** selectively evicts KV pairs based on the importance score while always preserving as many text features as the budget allows.

observation window (Li et al., 2024b), sum of self-attention, sum of intra-modality attention (VV+TT), and combination of intra- and inter- modality attention (TV+TT).

The results are presented in **block (b)** of Table 1, indicating Observation 3.2. This motivates us to design a robust KV importance criteria for VLM.

We show the results of randomly evicting text and vision KV pairs in **block (c)** in Table 1. It is worth noting that eviction of a small subset of text KV pairs cause much more damage to a larger subset of vision KV pairs, calling for extra protection to text KV pairs during eviction.

**Observation 3.2.** Integrating inter-modality interaction (TV) with intra-modality attention (TT) yields the most robust importance metric, indicating that visual saliency is intrinsically text-dependent.

**Observation 3.3.** Text features are highly sensitive to eviction, causing performance collapse, whereas visual features exhibit high redundancy, permitting aggressive eviction.

### 3.3. TGV-KV

**Text-Vision Budgeting (TVB).** The VLM handles different information levels in a hierarchical way (Liu et al., 2025a), and different budgets should be assigned to different layers (Qin et al., 2025). Instead of using a fixed rule-based policy (Cai et al., 2024) or a calibration dataset (Wang et al., 2025a;b), we adopt a cross-attention-driven policy to perform on-the-fly budgeting.

Based on Observation 3.1, we first slice text-vision attention $\mathbf{A}_l^{(TV)} \in \mathbb{R}^{N_t \times N_v}$ out of $\mathbf{A}_l$, which is calculated by

$$\mathbf{A}_l^{(TV)} = \text{softmax}(\mathbf{Q}_l^{(T)}[\mathbf{K}_l^{(V)}]^T) \in \mathbb{R}^{N_t \times N_v}, \quad (6)$$

where $\mathbf{Q}_l^{(T)} \in \mathbb{R}^{N_t \times d}$ and $\mathbf{K}_l^{(V)} \in \mathbb{R}^{N_v \times d}$ denotes the text query and vision key in layer $l$. We then sum up the text-vision attention in each layer and normalize along the layer to get the budget $b_l$ by Eq. 5.

**Text-Weighted Ranking (TWR).** We assign an importance ratio to each vision token and rank them in TWR. As shown in **set (b)** of Table 1, the sum of text-vision attention is ideal for visual KV eviction. However, this straightforward criterion lacks the inner importance of text tokens. For example, for instructions "Describe this image." and "Is there a taxi near the streetlight?", the text KV pairs are not equally important (Zhang et al., 2025c), and the preserved KV pairs should demonstrate different focuses. To this end, we use text tokens to weight vision tokens.

In Fig. 2, when zooming the text-text attention area, there exists a set of text tokens that consistently take up a large proportion of attention in all the subsequent tokens (Xiao et al., 2024; Qin et al., 2025), manifesting as a continuous prominent vertical line. We denote these text tokens as dominant text tokens. Since the dominant text tokens are critical for text understanding and generation, it is necessary to assign a higher importance score to vision KV pairs that are attended by dominant text tokens.

Specifically, we slice text-text attention $\mathbf{A}_l^{(TT)} \in \mathbb{R}^{N_t \times N_t}$ out of $\mathbf{A}_l$ and compute the column-wise sum of each text token. To account for the triangular causal mask, we divide each token's sum by the position to get an average

$$w_{l,j} = \frac{\sum_{i=j}^{N_t} [\mathbf{A}_l^{(TT)}]_{ij}}{N_t - j + 1}, j = 1 \cdots N_t. \quad (7)$$

Upon computing the significance score for each text KV

pair, we reuse $\mathbf{A}_l^{(\text{TV})}$ in Eq. 6 and multiply each $i$-th row with the normalized weight $\tilde{w}_{l,i}$ to get a text-weighted cross-modality attention summation. We compute the final importance score $s_{l,j}^{(\text{V})}$ for each vision KV in layer $l$ by

$$s_{l,j}^{(\text{V})} = \sum_{i=1}^{N_t} \tilde{w}_{l,i}[\mathbf{A}_l^{(\text{TV})}]_{ij}, j = 1, \cdots, N_v. \quad (8)$$

For text KVs, we directly take the sum of self-attention score as the importance score $s_{l,j}^{(\text{T})}$ since it has been proven to be effective and widely adopted (Zhang et al., 2023; Feng et al., 2024), which is calculated by

$$s_{l,j}^{(\text{T})} = \sum_{i=j}^{N_t} [\mathbf{A}_l^{(\text{TT})}]_{ij}, j = 1, \cdots, N_t. \quad (9)$$

**Text-Prioritised Retention (TPR).** Similar to KV cache eviction, most pruning methods (Chen et al., 2024a; Yang et al., 2025b; Liu et al., 2025a) prune vision tokens only since pruning text tokens often leads to extreme damage to model performance. As shown in our Observation 3.3, pruning text tokens significantly damages the performance. To mitigate this issue, we always try to keep as many text KV pairs as budget allows. We use a text-prioritised retention criterion, which only evicts text KVs when the retained exceeds the budget, though all the vision KVs are removed.

Let $\mathcal{T} = \{1, \cdots, N_t\}$ and $\mathcal{V} = \{1, \cdots, N_v\}$ denote the sets of indices for text and vision KV pairs, respectively. For layer $l$, the set of retained indices $\mathcal{I}_l$ is determined by:

$$\mathcal{I}_l = \begin{cases} \mathcal{T} \cup \text{TopK}(\{s_{l,j}^{(\text{V})}\}_{j\in\mathcal{V}}, b_l - N_t), & \text{if } b_l > N_t \\ \text{TopK}(\{s_{l,j}^{(\text{T})}\}_{j\in\mathcal{T}}, b_l). & \text{if } b_l \leq N_t \end{cases} \quad (10)$$

This criterion ensures that vision KVs are only retained after the entire text context is secured, and text KVs are only evicted under extreme budget constraints.

# 4. Experiments

## 4.1. Experiment Settings

**Models.** We evaluate TGV-KV on multiple VLMs with different architectures, including a basic model LLaVA-1.5-7B (Liu et al., 2024), high-resolution models LLaVA-NeXT-7B (Liu et al., 2024) and LLaVA-OV (Li et al., 2024a), and state-of-the-art open-source model Qwen3-VL-series with various sizes of 4B and 8B (Bai et al., 2025). All the models are evaluated without finetuning.

**Datasets.** We evaluate TGV-KV on both image and video tasks. For image tasks, we choose two types of tasks where text or vision dominates to get a comprehensive evaluation

of the model's ability. Vision-dominant tasks require the model to observe the image carefully, and we choose four representative VQA tasks, including ChartQA (Masry et al., 2022), DocVQA (Mathew et al., 2021), VizWiz (Gurari et al., 2018), and TextVQA (Singh et al., 2019). Text-dominant tasks mainly focus on the model's generation quality based on the observation, and we choose TextCaps (Sidorov et al., 2020) and COCO-Caption-2017 (Lin et al., 2015) for this type. For video tasks, we adopt Video-TT (Zhang et al., 2025b), which comprises rephrased, wrongly-led, and correctly-led adversarial open-ended questions to evaluate reasoning ability and robustness. More details on dataset description can be found in **Appendix** A.1.

**Comparisons.** We compare TGV-KV with multiple KV eviction methods, including those originally designed for LLMs (StreamingLLM (Xiao et al., 2024), SnapKV (Li et al., 2024b), H$_2$O (Zhang et al., 2023)) and VLMs (ElasticCache (Liu et al., 2025b), PrefixKV (Wang et al., 2025a)). Among them, H$_2$O and PrefixKV mainly adopt the sum of self-attention to allocate budget or evict KV pairs, SnapKV utilizes an observation window to assign importance score, and StreamingLLM preserves KV of the first and latest tokens that receive most attention. Since most results on new models are not reported by the original paper, we follow the original procedure and reproduce all the results ourselves. We set the observation window to 64 for SnapKV and the sink token number to 4 for StreamingLLM.

**Implementation Details.** We conduct all the accuracy evaluations with the LMMs-Eval toolkit (Zhang et al., 2024). All the experiments for image tasks are performed on a machine with $4\times$ RTX 5090 (32G), while video tasks are implemented on a machine with $4\times$ A800 (80G). Please refer to **Appendix** A.2 for more implementation details.

## 4.2. Accuracy Results

**Vision-Dominant Results.** We show the results for vision-dominant tasks in Table 2. Notably, TGV-KV establishes a new state-of-the-art across nearly all tasks and KV budgets.

We utilize LLaVA as a representative baseline model due to its straightforward architecture. Unlike modern VLMs with sufficient training and optimization, LLaVA is highly susceptible to error evictions, and any suboptimal token eviction leads to severe performance degradation. We show that most methods struggle with this model with an extreme budget of 5%, with some performance metrics plummeting to as low as 5% of the original performance. In contrast, TGV-KV consistently outperforms these baselines. On LLaVA-NeXT, a model with a higher resolution and a longer input sequence, TGV-KV reserves **99.2%** accuracy on VizWiz, and **97.4%** on TextVQA with only **5%** of the original KV cache size. These results highlight TGV-KV's

*Table 2.* **Accuracy Results on Vision-Dominant Tasks.** *Vanilla* refers to the baseline model using the full KV cache. The percentages indicate the specific KV cache retention rates. The best results in each setting are highlighted in **bold**.

| Methods | ChartQA[Relaxed Acc.↑] | | | | DocVQA[ANLS↑] | | | | VizWiz[Acc.↑] | | | | TextVQA[Acc.↑] | | | |
|---|---|---|---|---|---|---|---|---|---|---|---|---|---|---|---|---|
| | 50% | 20% | 10% | 5% | 50% | 20% | 10% | 5% | 50% | 20% | 10% | 5% | 50% | 20% | 10% | 5% |
| *LLaVA-1.5-7B (Liu et al., 2024)* | | | | | | | | | | | | | | | | |
| Vanilla | 18.0 | 18.0 | 18.0 | 18.0 | 23.9 | 23.9 | 23.9 | 23.9 | 54.4 | 54.4 | 54.4 | 54.4 | 47.9 | 47.9 | 47.9 | 47.9 |
| StreamingLLM | 15.2 | 14.4 | 14.3 | 13.4 | 17.8 | 15.3 | 14.4 | 13.6 | 53.0 | 52.5 | 52.1 | **48.2** | 40.5 | 35.4 | 33.8 | 32.5 |
| SnapKV | **17.9** | 16.6 | 0.4 | 0.4 | 23.0 | 20.5 | 1.7 | 1.7 | 54.1 | 53.6 | 6.5 | 5.0 | 47.3 | 44.6 | 9.2 | 9.2 |
| H$_2$O | 4.6 | 3.7 | 3.7 | 3.7 | 13.5 | 7.4 | 5.9 | 4.5 | 13.6 | 10.0 | 8.9 | 7.6 | 34.3 | 27.6 | 24.7 | 21.0 |
| ElasticCache | 2.6 | 3.2 | 3.4 | 3.9 | 6.2 | 3.7 | 3.6 | 2.8 | 4.7 | 6.1 | 6.5 | 9.3 | 20.8 | 14.6 | 13.1 | 9.1 |
| PrefixKV | 16.0 | 2.0 | 1.0 | 0.9 | 23.5 | 7.5 | 2.9 | 1.6 | 22.5 | 3.9 | 1.4 | 0.5 | 47.5 | 19.6 | 12.4 | 4.8 |
| **TGV-KV** | **17.9** | **18.0** | **15.5** | **13.8** | **23.6** | **21.8** | **19.2** | **14.3** | **54.4** | **54.0** | **52.3** | 31.2 | **47.8** | **47.4** | **44.5** | **33.9** |
| *LLaVA-NeXT-Mistral-7B (Liu et al., 2024)* | | | | | | | | | | | | | | | | |
| Vanilla | 52.9 | 52.9 | 52.9 | 52.9 | 63.7 | 63.7 | 63.7 | 63.7 | 63.7 | 63.7 | 63.7 | 63.7 | 65.7 | 65.7 | 65.7 | 65.7 |
| StreamingLLM | 39.0 | 30.0 | 29.7 | 30.0 | 46.3 | 33.2 | 29.5 | 28.1 | 62.4 | 60.8 | 60.5 | 60.5 | 58.0 | 47.4 | 44.6 | 43.1 |
| SnapKV | 52.4 | 49.6 | 46.2 | 39.2 | 63.3 | 60.8 | 55.1 | 44.0 | 63.5 | 63.0 | 62.5 | 61.5 | 65.3 | 63.0 | 59.8 | 54.9 |
| H$_2$O | 50.1 | 34.5 | 27.2 | 27.8 | 62.9 | 44.2 | 35.1 | 29.4 | 52.0 | 26.5 | 24.2 | 24.0 | 65.1 | 60.4 | 53.6 | 48.7 |
| ElasticCache | 44.5 | 31.1 | 22.7 | 18.5 | 57.9 | 40.3 | 27.8 | 19.9 | 26.3 | 24.2 | 20.4 | 18.9 | 61.7 | 47.3 | 35.5 | 31.7 |
| PrefixKV | 52.2 | 38.0 | 12.6 | 10.4 | 62.6 | 38.3 | 18.8 | 16.3 | 63.5 | 62.8 | 52.5 | 19.2 | 65.1 | 58.7 | 39.9 | 34.5 |
| **TGV-KV** | **52.6** | **51.8** | **51.2** | **49.3** | **63.7** | **62.8** | **61.4** | **58.5** | **63.7** | **63.6** | **63.6** | **63.2** | **65.7** | **65.5** | **65.1** | **64.0** |
| *LLaVA-OneVision-Qwen2-0.5B (Li et al., 2024a)* | | | | | | | | | | | | | | | | |
| Vanilla | 59.0 | 59.0 | 59.0 | 59.0 | 61.9 | 61.9 | 61.9 | 61.9 | 47.4 | 47.4 | 47.4 | 47.4 | 64.8 | 64.8 | 64.8 | 64.8 |
| StreamingLLM | 37.7 | 28.2 | 26.5 | 25.8 | 39.4 | 28.2 | 24.4 | 23.1 | 43.8 | 42.1 | 41.7 | 34.3 | 51.5 | 39.9 | 36.3 | 41.4 |
| SnapKV | 53.1 | 49.6 | 41.8 | 30.4 | 52.2 | 47.3 | 39.7 | 30.0 | 45.5 | 44.2 | 43.0 | 42.0 | 60.4 | 54.3 | 47.5 | 40.1 |
| H$_2$O | 16.8 | 37.8 | 33.4 | 31.0 | 21.5 | 41.2 | 31.0 | 20.7 | 19.5 | 34.0 | 31.3 | 30.0 | 19.2 | 56.0 | 51.7 | 43.2 |
| ElasticCache | 29.8 | 21.0 | 13.4 | 9.8 | 19.1 | 12.4 | 9.7 | 8.0 | 29.4 | 29.7 | 32.8 | 21.4 | 41.9 | 35.2 | 27.9 | 31.3 |
| PrefixKV | 14.0 | 42.7 | 24.6 | 13.7 | 35.9 | 25.9 | 16.5 | 8.8 | 23.3 | 28.8 | 29.2 | 30.9 | 40.7 | 48.4 | 31.4 | 23.6 |
| **TGV-KV** | **53.5** | **50.7** | **47.6** | **42.5** | **52.5** | **48.5** | **42.8** | **36.4** | **45.9** | **44.7** | **43.7** | **43.1** | **61.5** | **57.0** | **52.0** | **46.0** |
| *Qwen3-VL-4B-Instruct (Bai et al., 2025)* | | | | | | | | | | | | | | | | |
| Vanilla | 84.1 | 84.1 | 84.1 | 84.1 | 93.6 | 93.6 | 93.6 | 93.6 | 68.5 | 68.5 | 68.5 | 68.5 | 80.6 | 80.6 | 80.6 | 80.6 |
| StreamingLLM | 79.1 | 70.6 | 67.4 | 66.9 | 76.0 | 62.2 | 51.9 | 47.0 | 66.5 | 63.8 | 61.5 | 59.6 | 72.3 | 63.8 | 56.5 | 51.7 |
| SnapKV | 83.1 | 74.7 | 23.5 | 19.8 | 93.5 | 90.9 | 76.7 | 12.3 | 68.0 | 64.9 | 62.1 | 48.4 | 80.3 | 75.8 | 58.1 | 27.0 |
| H$_2$O | 82.4 | 71.0 | 65.8 | 63.4 | 92.7 | 82.9 | 67.7 | 53.6 | 67.5 | 64.2 | 62.6 | 54.8 | 80.2 | 74.5 | 65.2 | 55.7 |
| ElasticCache | 80.7 | 74.6 | 68.1 | 58.8 | 78.6 | 57.8 | 42.9 | 33.1 | 55.0 | 54.7 | 54.0 | 52.8 | 67.7 | 57.4 | 47.0 | 39.8 |
| PrefixKV | 83.4 | 78.1 | 72.9 | 65.4 | 92.2 | 73.1 | 52.8 | 43.0 | 67.2 | 59.2 | 55.5 | 49.5 | 79.9 | 67.9 | 53.0 | 47.6 |
| **TGV-KV** | **83.8** | **82.5** | **80.2** | **67.2** | **93.5** | **93.1** | **91.9** | **87.3** | **68.4** | **67.8** | **66.2** | **63.1** | **80.6** | **80.1** | **78.1** | **70.4** |
| *Qwen3-VL-8B-Instruct (Bai et al., 2025)* | | | | | | | | | | | | | | | | |
| Vanilla | 85.3 | 85.3 | 85.3 | 85.3 | 95.1 | 95.1 | 95.1 | 95.1 | 70.0 | 70.0 | 70.0 | 70.0 | 82.1 | 82.1 | 82.1 | 82.1 |
| StreamingLLM | 80.8 | 73.8 | 70.8 | 72.0 | 81.2 | 64.8 | 54.6 | 49.6 | 69.2 | 66.5 | 63.5 | 61.3 | 76.9 | 68.7 | 61.2 | 54.0 |
| SnapKV | 85.0 | 78.1 | 42.3 | 39.3 | 94.9 | 91.2 | 80.1 | 22.6 | 69.8 | 67.4 | 64.9 | 53.8 | 81.9 | 78.6 | 64.0 | 40.2 |
| H$_2$O | 84.0 | 78.0 | 72.7 | 67.6 | 94.4 | 86.5 | 72.0 | 58.1 | 69.7 | 67.6 | 62.0 | 56.6 | 81.6 | 77.5 | 66.7 | 58.5 |
| ElasticCache | 78.8 | 68.5 | 67.0 | 67.7 | 87.1 | 71.4 | 57.0 | 45.3 | 48.7 | 51.7 | 53.4 | 53.6 | 70.0 | 64.7 | 57.2 | 48.1 |
| PrefixKV | 84.7 | 78.9 | 74.2 | 71.2 | 93.9 | 72.7 | 50.3 | 45.4 | 68.8 | 59.7 | 57.1 | 55.7 | 81.4 | 69.3 | 54.0 | 49.1 |
| **TGV-KV** | **85.3** | **84.4** | **82.1** | **73.1** | **95.0** | **94.7** | **93.2** | **88.0** | **69.8** | **69.2** | **67.6** | **64.4** | **82.1** | **81.8** | **79.4** | **72.0** |

robustness under high-resolution scenarios.

Beyond the LLaVA series, we evaluate Qwen3-VL, which represents the state-of-the-art in multimodal understanding. Specifically, with a budget of **5%**, TGV-KV preserves **93.3%** performance on DocVQA, and **92.1%** on VizWiz on Qwen3-VL-4B. On a larger version with 8B parameters, TGV-KV also achieves superior results, showcasing outstanding scaling ability to models with different sizes.

**Text-Dominant Results.** Results for image tasks where text dominates are presented in Table 3. On TextCaps, TGV-KV maintains the performance decrease **less than 0.5%** with both LLaVA and Qwen under a retained budget of **50%**. It is notable that most existing methods suffer from

catastrophic performance degradation when the budget is strictly limited, while TGV-KV still maintains accuracy under these scenarios, significantly outperforming the second-best method by a large margin, gaining a **57.4%** relative performance boost compared with StreamingLLM with LLaVA. These results highlight TGV-KV's ability in text-dominant tasks and resource-limited devices.

**Video Results.** We plot the results on video tasks in the form of Pareto curves in Fig 4. Video-TT mainly evaluates VLM's reasoning ability and robustness under misleading instructions, which incorporate long context generation and LLM-as-a-Judge assessing. Following the trend in image tasks, TGV-KV maintains performance comparable to the

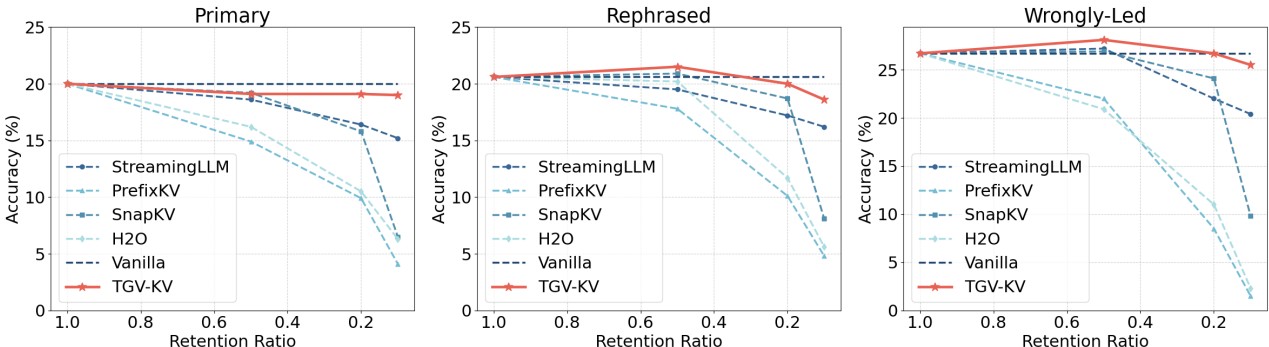

*Figure 4.* **Accuracy Results on Video-TT (Zhang et al., 2025b)** using Qwen3-VL-4B. *Vanilla* refers to the original model with the full computation budget. Detailed breakdowns on subtasks and full numerical results are provided in **Appendix** B.2.

*Table 3.* **Accuracy Results on Text-Dominant Tasks.** The results of *CIDEr* and *ROUGE-L* are shown in percentage. The full version with more metrics can be found in **Appendix** B.1.

| Methods | TextCaps$^{CIDEr}\uparrow$ | | | COCO-Cap$^{ROUGE-L}\uparrow$ | | |
|---|---|---|---|---|---|---|
| | 50% | 20% | 10% | 50% | 20% | 10% |
| *LLaVA-1.5-7B (Liu et al., 2024)* | | | | | | |
| Vanilla | 100.3 | 100.3 | 100.3 | 55.1 | 55.1 | 55.1 |
| StreamingLLM | 78.4 | 53.3 | 40.4 | 54.3 | 52.2 | 51.0 |
| SnapKV | 97.1 | 82.3 | 0.3 | 55.2 | 54.6 | 0.9 |
| $H_2O$ | 14.3 | 2.4 | 0.6 | 17.7 | 9.8 | 8.0 |
| ElasticCache | 3.3 | 1.3 | 1.1 | 9.6 | 8.7 | 8.4 |
| PrefixKV | 94.3 | 12.0 | 1.4 | 34.0 | 11.1 | 7.9 |
| **TGV-KV** | **99.8** | **87.8** | **63.6** | **55.3** | **55.0** | **52.6** |
| *Qwen3-VL-8B-Instruct (Bai et al., 2025)* | | | | | | |
| Vanilla | 33.6 | 33.6 | 33.6 | 42.0 | 42.0 | 42.0 |
| StreamingLLM | **34.9** | 32.2 | 24.5 | 41.7 | 39.6 | 35.3 |
| SnapKV | 34.1 | 34.4 | 27.2 | 40.4 | 36.3 | 35.7 |
| $H_2O$ | 32.3 | 34.1 | 28.8 | 41.4 | 39.3 | 37.9 |
| ElasticCache | 27.4 | 26.6 | 20.1 | 39.9 | 38.1 | 34.6 |
| PrefixKV | 32.6 | 27.7 | 14.4 | 41.7 | 40.6 | 36.2 |
| **TGV-KV** | 33.5 | **34.9** | **30.9** | **42.0** | **41.6** | **39.5** |

*Table 4.* **Memory and Speed Analyses.** Evaluated with LLaVA-1.5-7B on an NVIDIA A800 GPU. *Memory* refers to the KV cache consumption. *Latency* denotes the average time per decoding step. *Throughput* accounts for both prefill and decoding stages.

| Settings | Memory (GB)↓ | Latency (ms)↓ | Throughput (tokens/s)↑ |
|---|---|---|---|
| *Context Length=8k tokens* | | | |
| Vanilla-`Eager` | 3.91 | 40.1±1.5 | 23.6±0.1 |
| Vanilla-`FA2` | 3.91 | 44.1±0.9 | 25.3±0.5 |
| TGV-KV/50% | 1.95 | 29.4±0.4 (-26.7%) | 26.2±0.4 (+11.0%) |
| TGV-KV/10% | 0.39 | 30.9±3.5 (-22.9%) | 29.8±0.3 (+26.3%) |
| TGV-KV/5% | 0.20 | 27.9±0.4 (-30.4%) | 31.0±0.1 (+31.4%) |
| *Context Length=16k tokens* | | | |
| Vanilla-`Eager` | 7.81 | 57.3±0.7 | 21.1±0.3 |
| Vanilla-`FA2` | 7.81 | 57.4±0.3 | 21.3±0.3 |
| TGV-KV/50% | 3.91 | 33.6±1.2 (-41.4%) | 24.1±0.1 (+14.2%) |
| TGV-KV/10% | 0.78 | 28.1±0.1 (-51.0%) | 30.8±0.3 (+46.0%) |
| TGV-KV/5% | 0.39 | 27.9±0.4 (-51.3%) | 32.2±0.3 (+52.6%) |

vanilla model baseline even as the retention ratio decreases sharply. In the high-compression setting where only **10%** of the vanilla budget is retained, TGV-KV only decreases $\leq$**2** percentage points on Rephased and Wrongly-Led instructions. On the primary subtask, TGV-KV consistently maintains over **95%** of the original performance and stays close to the vanilla model under all the budgets.

### 4.3. Efficiency Results

**Memory and Speed.** We evaluate the computational efficiency of TGV-KV with different context lengths, with results on memory, per-token latency, and throughput in Table 4. Notably, with an extreme retention rate of 5%, TGV-KV reduces the memory consumption of KV cache size from 3.91 GB to a mere 0.20 GB under an 8k context. This substantial reduction effectively alleviates the memory bottleneck often encountered during the deployment

of VLM. Furthermore, while FlashAttention (Dao, 2024) significantly accelerates the prefill stage by optimizing full-sequence attention computation, it provides limited or even negative speedup during decode due to the inherently low parallelism of single-token queries. These results showcase the efficiency of TGV-KV towards long context generation.

### 4.4. Ablation Studies

To systematically validate the effectiveness of each component, we conduct ablation studies by evaluating all combinations of our three proposed modules. As reported in Table 5, we can draw three key conclusions on the role of each component. **(1)** TWR consistently yields superior importance criteria, as almost all the sets utilizing TWR for the importance score surpass those simply using naive self-attention. **(2)** TPR is crucial under extreme KV cache budgets, since TPR brings much better results under 5% budget, especially when TWR is absent. **(3)** TVB provides supplementary performance gains when the budget is sufficient, assisting in allocating extra budget to key layers. The ablation stud-

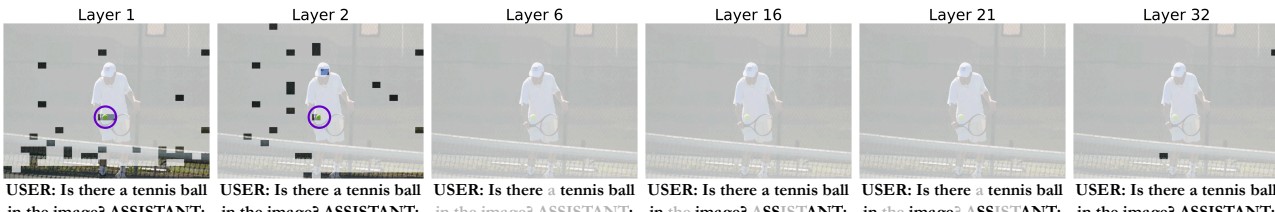

*Figure 5.* **Visualization of Retained KV.** We plot the retained KV of one sample from POPE (Li et al., 2023). The visualization is from LLaVA (32 layers) with an overall retention of 3%. The patches most related to the text instruction are circled in violet. Evicted KVs are masked or labelled in gray. More visualizations are in **Appendix** C.2.

*Table 5.* **Ablation Studies.** Results are carried out with LLaVA-1.5-7B. *Importance Score* stands for the criterion to judge KV pair importance, and tokens with lower scores are first to be evicted. Ablation studies inside each submodule are presented in Table 1.

| TVB | TWR | TPR | ChartQA↑ | | COCO-Cap↑ | |
|---|---|---|---|---|---|---|
| | | | 20% | 5% | 20% | 5% |
| *Attention Sum → Importance Score* | | | | | | |
| | | | 3.8 (-14.2) | 4.5 (-9.3) | 20.0 (-35.0) | 18.8 (-30.3) |
| | | ✓ | 17.4 (-0.6) | 13.6 (-0.2) | 54.8 (-0.2) | 47.9 (-1.4) |
| ✓ | | | 15.0 (-3.0) | 1.1 (-12.7) | 54.9 (-0.1) | 13.4 (-35.7) |
| ✓ | | ✓ | 17.3 (-0.7) | 13.0 (-0.8) | 54.8 (-0.2) | 48.9 (-0.2) |
| *Text-Weighted Ranking → Importance Score* | | | | | | |
| | ✓ | | 17.8 (-0.2) | 13.5 (-0.3) | 55.1 (+0.1) | 48.6 (-0.5) |
| | ✓ | ✓ | 17.8 (-0.2) | 13.8 (-0.0) | 55.1 (+0.1) | 49.1 (-0.0) |
| ✓ | ✓ | | 18.0 (-0.0) | 13.7 (-0.1) | 55.0 (-0.0) | 46.0 (-3.1) |
| ✓ | ✓ | ✓ | 18.0 | 13.8 | 55.0 | 49.1 |

*Table 6.* **Accuracy Results on Token Pruning and KV Eviction.** We evaluate TGV-KV against recent token pruning methods. The number denotes the equivalent retained token or KV size.

| Methods | MME↑ | | | POPE↑ | | | GQA↑ | | |
|---|---|---|---|---|---|---|---|---|---|
| | 128 | 64 | 32 | 128 | 64 | 32 | 128 | 64 | 32 |
| Vanilla | 1781 | 1781 | 1781 | 84.6 | 84.6 | 84.6 | 60.7 | 60.7 | 60.7 |
| DivPrune | 1724 | 1636 | 1600 | 86.9 | 85.7 | 81.4 | 59.2 | 57.5 | 55.3 |
| VisionZip | 1755 | 1693 | 1585 | 83.1 | 76.9 | 70.4 | 57.6 | 55.1 | 52.3 |
| VisPruner | 1766 | 1696 | 1552 | 84.5 | 78.3 | 73.3 | 58.5 | 55.3 | 52.8 |
| CDPruner | 1746 | 1707 | 1692 | 87.0 | 87.0 | 87.5 | 59.8 | 58.8 | 57.4 |
| **TGV-KV** | 1781 | 1781 | 1781 | 84.6 | 84.6 | 84.6 | 60.7 | 60.6 | 60.4 |

ies inside each submodule have been discussed in Table 1. These results corroborate the design philosophy of TGV-KV, underscoring the necessity of tailored strategies to bridge the modality gap in VLMs, rather than a straightforward transplantation of prevailing LLM paradigms.

### 4.5. Discussions

**Why Compress KV Rather Than Token?** We prioritize KV cache eviction over token pruning for two main reasons. **(1)** Superior accuracy under the same budget. When evicting a token's KV in one certain layer, its information is still visible in subsequent layers where the corresponding KV is not evicted. However, once a token is pruned, its unextracted information can no longer be accessed. As shown in Table 6, under a same budget, TGV-KV barely degrades the VQA accuracy or increases hallucination. **(2)** System efficiency gains. Although token pruning reduces computation in both prefill and decoding, KV eviction mainly accelerates decoding. The overall latency is dominated by the iterative decoding stage rather than the one-time prefill computation (Liu et al., 2025b). Consequently, token pruning offers limited practical acceleration while significantly harming model quality. TGV-KV provides a more favorable trade-off between efficiency and performance in deployment.

**Visualizations of Evicted KVs.** We visualize the retained text and vision KV in different layers in Fig. 5. TVB tends to assign more budget to shallow layers, indicating that cross-modality interaction is intense in these layers, aligning with previous studies (Chen et al., 2024a; Xing et al., 2025). In some middle layers with very limited budget, all the vision KV and a few text KV are evicted, while the key part of the text instruction, e.g., "tennis ball" in this example, is reserved across all the layers. Besides, the image patches most related to these dominant text tokens are also accurately preserved in shallow layers. These findings support the rationale behind TWR and unveil its potential in visual grounding and hallucination alleviation.

## 5. Conclusion

In this paper, we take a systematic study of the multimodal attention pattern in VLMs and concludes three key observations vital to multimodal KV cache eviction design. Based on the analyses, we overcome the modality gap in VLM and propose a robust KV cache eviction approach, TGV-KV, which fully leverages the text to guide vision KV eviction. TGV-KV consists of three modules, where TVB allocates layer-wise budgets, TWR evaluates the KV importance score, and TPR preserves crucial text information. We evaluate TGV-KV across multiple models and benchmarks, proving its effectiveness in multimodal KV eviction. Our conclusions and observations are universal, and we believe subsequent researches can draw inspiration from our study.

## Acknowledgement

This work is partially sponsored by the National Natural Science Foundation of China under Grant 62306084 and U23B2051, Shenzhen College Stability Support Plan under Grant GXWD20231128102243003, and Shenzhen Science and Technology Program under Grant ZDSYS20230626091203008 and KJZD20230923115113026.

## Impact Statement

This paper presents work whose goal is to advance the field of Machine Learning. There are many potential societal consequences of our work, none which we feel must be specifically highlighted here.

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

# A. Experiment Details

## A.1. Dataset Description

### A.1.1. IMAGE TASKS

**Vision-Dominant Tasks.**

- **ChartQA** (Masry et al., 2022). ChartQA is a QA dataset on charts. It mainly evaluates the model's ability in visual feature extraction and logical reasoning. It contains two subsets, i.e., human-authored and machine-generated. Human-authored subset includes ∼9.6k high-quality human-written QA pairs, covering ∼4.8k charts. Machine-generated subset uses a T5 model to generate ∼23k QA pairs for ∼17k charts based on human-written chart captions. In our evaluation, we adopt both subsets and average them to get an overall score.

- **DocVQA** (Mathew et al., 2021). The main goal of DocVQA is to test the model's capability in understanding visual cues, identifying text in different format and extractive QA. The whole dataset contains ∼12k images and ∼50k QA pairs. We evaluate on the validation set with ∼1.2k images and ∼5k QA pairs. We report the results in Average Normalized Levenshtein Similarity (ANLS), a metric designed for QA tasks.

- **VizWiz** (Gurari et al., 2018). VizWiz is a VQA task constructed with true questions and images from blind people. Beyond the traditional VQA task, VizWiz further includes an answerability prediction, which requires the model to first judge whether the question can be answered, since some images are blurry or have no meaningful objects. The whole dataset includes ∼31k items, and we evaluate on the validation set with a size of ∼3.1k.

- **TextVQA** (Singh et al., 2019). TextVQA tests the model in reading image and comprehension. Common VQA tasks feature relatively fewer questions on reading the image, while TextVQA challenges include more OCR and reasoning ability. To answer the questions, the model needs to locate the related text and decide whether directly use the seen text as an answer or think before answering. The whole dataset features ∼45k questions, and we adopt the 5k validation set.

**Text-Dominant Tasks.**

- **TextCaps** (Sidorov et al., 2020). TextCaps is a caption challenge that tests the model in reading text in the image and generating continuous text, which requires OCR capability, text understanding, and paraphrasing ability. It contains ∼28k images and ∼145k captions, and we conduct all the evaluation on the validation set, with ∼3.1k items. It uses multiple metrics, including Bleu, METEOR, ROUGE-L, and CIDEr. We report the results in CIDEr in our main paper.

- **COCO-Caption** (Lin et al., 2015). COCO dataset is a large-scale computer vision dataset that comprises object detection, instance segmentation, semantic segmentation, image caption, and so on. In our study, we mainly evaluate the model's ability in text-dominant tasks and only use the caption task. We utilize the validation set from COCO-2017, which features 5k data items on captioning. We report ROUGE-L in the main paper.

### A.1.2. VIDEO TASKS

- **Video-TT** (Zhang et al., 2025b). Video-TT mainly evaluates the model in two aspects, robustness and correctness. There are 1k videos selected from YouTube Shorts within 65 seconds, paired with one primary open-ended question and four adversarial questions. The adversarial questions include rephrased questions, questions with correct leads, questions with wrong leads, and multiple-choice questions. Most of the videos include complex visual transformation or story plot, thus making the benchmark challenging even for closed-sourced commercial models. The answers are sent to LLM-as-a-Judge to get a final score.

## A.2. Implementation Details

The max vision tokens for Qwen3-VL is set to 1024, and the vision aspect ratio is set to 2 for LLaVA-OV due to several extremely large images in the dataset, which may cause Out-of-Memory during evaluation. Following the common practice, we retain the first 4 and last 1 tokens for all the methods to avoid performance collision. We also manually set the max generated tokens to 32 for all the tasks, which is longer than any answer response, to shorten the evaluation time, because some methods cause the loss of `EOS` token and the generation falls into a dead loop. For Video-TT, we use DeepSeek-V3.2 with thinking mode off for LLM-as-a-Judge evaluation.

*Table 7.* **Full performance comparison on text-dominant benchmarks.** We report BLEU-x (B-x), CIDEr (C), METEOR (M), and ROUGE-L (R). Note that CIDEr scores are reported as raw values.

| Methods | Retain 50% KV | | | | | | | Retain 20% KV | | | | | | | Retain 10% KV | | | | | | |
|---|---|---|---|---|---|---|---|---|---|---|---|---|---|---|---|---|---|---|---|---|---|
| | B-1 | B-2 | B-3 | B-4 | C | M | R | B-1 | B-2 | B-3 | B-4 | C | M | R | B-1 | B-2 | B-3 | B-4 | C | M | R |
| *TextCaps (Sidorov et al., 2020) w/ LLaVA-1.5 (Liu et al., 2024)* | | | | | | | | | | | | | | | | | | | | | |
| Vanilla | 0.71 | 0.53 | 0.38 | 0.27 | 1.00 | 0.23 | 0.47 | 0.71 | 0.53 | 0.38 | 0.27 | 1.00 | 0.23 | 0.47 | 0.71 | 0.53 | 0.38 | 0.27 | 1.00 | 0.23 | 0.47 |
| StreamingLLM | 0.68 | 0.49 | 0.34 | 0.24 | 0.78 | 0.21 | 0.44 | 0.64 | 0.44 | 0.30 | 0.19 | 0.53 | 0.19 | 0.41 | 0.61 | 0.41 | 0.26 | 0.17 | 0.40 | 0.17 | 0.39 |
| SnapKV | **0.71** | 0.52 | 0.37 | 0.26 | 0.97 | **0.23** | **0.47** | 0.68 | 0.49 | 0.34 | 0.24 | 0.82 | **0.22** | **0.45** | 0.00 | 0.00 | 0.00 | 0.00 | 0.00 | 0.01 | 0.01 |
| H2O | 0.08 | 0.05 | 0.03 | 0.02 | 0.14 | 0.08 | 0.17 | 0.04 | 0.02 | 0.01 | 0.00 | 0.02 | 0.04 | 0.09 | 0.03 | 0.02 | 0.01 | 0.00 | 0.01 | 0.04 | 0.07 |
| ElasticCache | 0.04 | 0.02 | 0.01 | 0.01 | 0.03 | 0.05 | 0.09 | 0.04 | 0.02 | 0.01 | 0.00 | 0.01 | 0.04 | 0.07 | 0.04 | 0.01 | 0.01 | 0.00 | 0.01 | 0.03 | 0.07 |
| PrefixKV | 0.70 | 0.52 | **0.38** | **0.27** | 0.94 | 0.22 | **0.47** | 0.06 | 0.04 | 0.02 | 0.01 | 0.12 | 0.06 | 0.14 | 0.04 | 0.02 | 0.01 | 0.00 | 0.01 | 0.04 | 0.08 |
| **TGV-KV** | **0.71** | **0.53** | **0.38** | **0.27** | **1.00** | **0.23** | **0.47** | **0.70** | **0.51** | **0.36** | **0.25** | **0.88** | **0.22** | **0.45** | **0.64** | **0.45** | **0.30** | **0.20** | **0.64** | **0.19** | **0.42** |
| *COCO-Caption (Lin et al., 2015) w/ LLaVA-1.5 (Liu et al., 2024)* | | | | | | | | | | | | | | | | | | | | | |
| Vanilla | 0.73 | 0.56 | 0.41 | 0.29 | 1.08 | 0.28 | 0.55 | 0.73 | 0.56 | 0.41 | 0.29 | 1.08 | 0.28 | 0.55 | 0.73 | 0.56 | 0.41 | 0.29 | 1.08 | 0.28 | 0.55 |
| StreamingLLM | 0.73 | 0.56 | 0.40 | 0.29 | 1.04 | 0.27 | 0.54 | 0.71 | 0.53 | 0.38 | 0.26 | 0.93 | 0.25 | 0.52 | 0.70 | 0.52 | 0.37 | 0.25 | 0.86 | 0.24 | 0.51 |
| SnapKV | **0.74** | **0.57** | **0.42** | **0.30** | **1.09** | **0.28** | **0.55** | 0.73 | 0.56 | **0.41** | **0.29** | 1.06 | 0.27 | **0.55** | 0.00 | 0.00 | 0.00 | 0.00 | 0.00 | 0.01 | 0.01 |
| H2O | 0.07 | 0.05 | 0.03 | 0.02 | 0.15 | 0.09 | 0.18 | 0.04 | 0.02 | 0.01 | 0.01 | 0.03 | 0.05 | 0.10 | 0.04 | 0.02 | 0.01 | 0.00 | 0.01 | 0.04 | 0.08 |
| ElasticCache | 0.04 | 0.03 | 0.01 | 0.01 | 0.03 | 0.05 | 0.10 | 0.04 | 0.02 | 0.01 | 0.00 | 0.01 | 0.04 | 0.09 | 0.04 | 0.02 | 0.01 | 0.00 | 0.01 | 0.04 | 0.08 |
| PrefixKV | 0.15 | 0.11 | 0.08 | 0.05 | 0.63 | 0.14 | 0.34 | 0.05 | 0.03 | 0.02 | 0.01 | 0.06 | 0.06 | 0.11 | 0.04 | 0.02 | 0.01 | 0.00 | 0.01 | 0.04 | 0.08 |
| **TGV-KV** | **0.74** | **0.57** | **0.42** | **0.30** | **1.09** | **0.28** | **0.55** | **0.74** | **0.57** | **0.41** | **0.29** | **1.07** | **0.28** | **0.55** | **0.71** | **0.54** | **0.39** | **0.27** | **0.95** | **0.26** | **0.53** |
| *TextCaps (Sidorov et al., 2020) w/ Qwen3-VL-8B-Instruct (Bai et al., 2025)* | | | | | | | | | | | | | | | | | | | | | |
| Vanilla | 0.47 | 0.30 | 0.20 | 0.14 | 0.34 | 0.25 | 0.39 | 0.47 | 0.30 | 0.20 | 0.14 | 0.34 | 0.25 | 0.39 | 0.47 | 0.30 | 0.20 | 0.14 | 0.34 | 0.25 | 0.39 |
| StreamingLLM | 0.46 | 0.29 | 0.19 | 0.12 | **0.35** | 0.24 | 0.38 | 0.44 | 0.27 | 0.17 | 0.11 | 0.32 | 0.21 | 0.36 | 0.40 | 0.23 | 0.14 | 0.08 | 0.25 | 0.18 | 0.33 |
| SnapKV | **0.47** | 0.29 | **0.20** | 0.13 | 0.34 | 0.24 | 0.38 | 0.45 | **0.29** | 0.18 | 0.11 | 0.34 | 0.22 | 0.36 | 0.40 | 0.24 | 0.14 | 0.09 | 0.27 | 0.17 | 0.31 |
| H2O | **0.47** | **0.30** | **0.20** | **0.14** | 0.32 | **0.25** | **0.39** | 0.44 | 0.28 | **0.19** | **0.14** | 0.34 | 0.23 | **0.39** | 0.40 | 0.23 | 0.14 | **0.10** | 0.29 | 0.19 | **0.36** |
| ElasticCache | 0.44 | 0.28 | 0.19 | 0.12 | 0.27 | 0.24 | 0.37 | 0.42 | 0.26 | 0.16 | 0.10 | 0.27 | 0.21 | 0.35 | 0.39 | 0.23 | 0.13 | 0.08 | 0.20 | 0.18 | 0.32 |
| PrefixKV | 0.46 | **0.30** | **0.20** | 0.13 | 0.33 | **0.25** | 0.38 | 0.42 | 0.26 | 0.16 | 0.10 | 0.28 | 0.22 | 0.35 | 0.35 | 0.20 | 0.11 | 0.06 | 0.14 | 0.16 | 0.30 |
| **TGV-KV** | **0.47** | **0.30** | **0.20** | **0.14** | 0.33 | **0.25** | **0.39** | **0.46** | **0.29** | **0.19** | 0.13 | **0.35** | **0.24** | 0.38 | **0.41** | **0.25** | **0.16** | **0.10** | **0.31** | **0.21** | **0.36** |
| *COCO-Caption (Lin et al., 2015) w/ Qwen3-VL-8B-Instruct (Bai et al., 2025)* | | | | | | | | | | | | | | | | | | | | | |
| Vanilla | 0.48 | 0.31 | 0.19 | 0.12 | 0.27 | 0.25 | 0.42 | 0.48 | 0.31 | 0.19 | 0.12 | 0.27 | 0.25 | 0.42 | 0.48 | 0.31 | 0.19 | 0.12 | 0.27 | 0.25 | 0.42 |
| StreamingLLM | **0.48** | **0.31** | **0.19** | 0.11 | **0.29** | **0.25** | **0.42** | 0.45 | 0.28 | 0.17 | 0.10 | 0.29 | 0.22 | 0.40 | 0.39 | 0.23 | 0.13 | 0.07 | 0.18 | 0.19 | 0.35 |
| SnapKV | 0.47 | 0.29 | 0.18 | 0.11 | **0.29** | 0.24 | 0.40 | 0.42 | 0.27 | 0.16 | 0.10 | 0.25 | 0.17 | 0.36 | 0.42 | **0.27** | 0.16 | **0.10** | 0.24 | 0.17 | 0.36 |
| H2O | 0.47 | 0.30 | 0.18 | **0.12** | 0.23 | **0.25** | 0.41 | 0.40 | 0.25 | 0.18 | 0.09 | 0.25 | 0.23 | 0.39 | **0.46** | 0.25 | 0.16 | **0.10** | 0.26 | 0.20 | 0.38 |
| ElasticCache | 0.45 | 0.28 | 0.17 | 0.10 | 0.21 | 0.24 | 0.40 | 0.42 | 0.26 | 0.16 | 0.09 | 0.23 | 0.22 | 0.38 | 0.38 | 0.22 | 0.13 | 0.07 | 0.16 | 0.19 | 0.35 |
| PrefixKV | **0.48** | 0.30 | **0.19** | **0.12** | 0.28 | **0.25** | **0.42** | 0.46 | 0.29 | 0.18 | **0.11** | **0.30** | 0.23 | 0.41 | 0.40 | 0.24 | 0.14 | 0.08 | 0.19 | 0.20 | 0.36 |
| **TGV-KV** | **0.48** | **0.31** | **0.19** | **0.12** | 0.27 | **0.25** | **0.42** | **0.47** | **0.30** | **0.19** | **0.11** | 0.29 | **0.25** | **0.42** | **0.44** | **0.27** | **0.17** | **0.10** | **0.27** | **0.22** | **0.40** |

*Table 8.* **Extended Accuracy Results on Visual KV Eviction Methods.** *Vanilla* denotes the vanilla model with full KV. The percentage denotes the vision KV budget retention ratio, while all the text KV are retained. The best result of each set is marked in **bold**.

| Methods | ChartQA[Relaxed Acc.↑] | | | | DocVQA[ANLS↑] | | | | VizWiz[Acc.↑] | | | | TextVQA[Acc.↑] | | | |
|---|---|---|---|---|---|---|---|---|---|---|---|---|---|---|---|---|
| | 50% | 20% | 10% | 5% | 50% | 20% | 10% | 5% | 50% | 20% | 10% | 5% | 50% | 20% | 10% | 5% |
| *LLaVA-1.5-7B (Liu et al., 2024)* | | | | | | | | | | | | | | | | |
| Vanilla | 18.0 | 18.0 | 18.0 | 18.0 | 23.9 | 23.9 | 23.9 | 23.9 | 54.4 | 54.4 | 54.4 | 54.4 | 47.9 | 47.9 | 47.9 | 47.9 |
| AirCache | 17.6 | 17.0 | 15.6 | 15.0 | 23.0 | 20.6 | 18.6 | 16.8 | 54.3 | 53.9 | 53.5 | 52.9 | 47.8 | 46.1 | 43.8 | 41.1 |
| **TGV-KV** | **17.8** | **18.0** | **17.6** | **16.5** | **23.7** | **22.6** | **21.2** | **19.7** | **54.4** | **54.2** | **53.9** | **53.5** | **48.0** | **47.7** | **47.0** | **45.7** |
| *Qwen3-VL-8B-Instruct (Bai et al., 2025)* | | | | | | | | | | | | | | | | |
| Vanilla | 85.3 | 85.3 | 85.3 | 85.3 | 95.1 | 95.1 | 95.1 | 95.1 | 70.0 | 70.0 | 70.0 | 70.0 | 82.1 | 82.1 | 82.1 | 82.1 |
| AirCache | 84.1 | 81.5 | 78.8 | 76.5 | 94.8 | 93.3 | 90.5 | 85.5 | 69.7 | 69.0 | 68.5 | 66.7 | 81.7 | 80.9 | 78.7 | 74.3 |
| **TGV-KV** | **85.3** | **84.9** | **84.2** | **82.8** | **95.0** | **94.8** | **94.0** | **92.3** | **70.0** | **69.7** | **69.1** | **68.1** | **82.1** | **81.9** | **81.1** | **79.2** |

*Table 9.* **Detailed Results on Video-TT.** We evaluate the primary task and three robustness tasks.

| Methods | Primary↑ | | | Correctly-Led↑ | | | Wrongly-Led↑ | | | Paraphrase↑ | | |
|---|---|---|---|---|---|---|---|---|---|---|---|---|
| | 50% | 20% | 10% | 50% | 20% | 10% | 50% | 20% | 10% | 50% | 20% | 10% |
| *Qwen3-VL-4B-Instruct* (Bai et al., 2025) | | | | | | | | | | | | |
| Vanilla | 20.0 | 20.0 | 20.0 | 24.8 | 24.8 | 24.8 | 26.7 | 26.7 | 26.7 | 20.6 | 20.6 | 20.6 |
| StreamingLLM | 18.6 | 16.4 | 15.2 | **24.8** | 25.5 | 25.1 | 27.2 | 22.0 | 20.4 | 19.5 | 17.2 | 16.2 |
| SnapKV | **19.2** | 15.8 | 6.5 | 24.6 | 24.1 | 24.3 | 26.9 | 24.1 | 9.8 | 20.9 | 18.7 | 8.1 |
| H$_2$O | 16.2 | 10.5 | 6.3 | 23.8 | 14.8 | 5.5 | 20.9 | 11.0 | 2.3 | 20.2 | 11.7 | 5.6 |
| PrefixKV | 14.9 | 9.9 | 4.1 | 22.4 | 13.9 | 3.0 | 22.0 | 8.5 | 1.5 | 17.8 | 10.1 | 4.8 |
| **TGV-KV** | 19.1 | **19.1** | **19.0** | 24.4 | **25.6** | **25.3** | **28.1** | **26.7** | **25.5** | **21.5** | **20.0** | **18.6** |

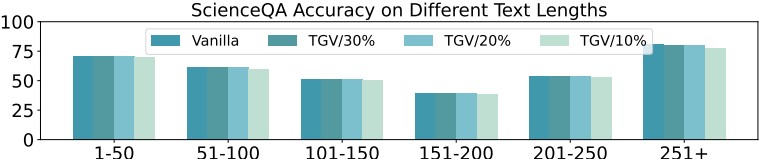

*Figure 6.* **Accuracy Results on Different Lengths.** We count the model accuracy with different text instruction lengths in ScienceQA (Lu et al., 2022). The results stay smooth and stable under different lengths, indicating removing all visual KV makes a small influence.

## B. Extended Experiment Results

### B.1. Text-Dominant Results

We provide a comprehensive evaluation of text-dominant tasks in Table 7, including more metrics (BLEU-1/2/3/4, CIDEr, METEOR, and ROUGE-L), which assess the model's ability in generating descriptive text based on visual inputs. An interesting phenomenon is that TGV-KV surpasses the vanilla model in several settings, suggesting that TGV-KV acts as an effective noise filter that evicts irrelevant information.

### B.2. Video Results

Video understanding poses a significant challenge for KV eviction due to the extended context length and temporal redundancy. We report the raw results of Video-TT (Zhang et al., 2025b) in Table 9, which evaluates reasoning capabilities under adversarial conditions. On the primary task, TGV-KV retains **95.0%** of the vanilla performance even when the budget is only **10%** of the full KV. On the wrongly-led subtask, which tests the model's ability to ignore hallucinatory instructions, TGV-KV achieves **95.5%** of the vanilla performance. This indicates that our Text-Weighted Ranking (TWR) successfully prioritizes visual frames and patches that are semantically grounded in the query, allowing the model to answer correctly despite adversarial leads.

### B.3. Results Against Visual KV Eviction Method

Some related works (Huang et al., 2025) also conduct KV eviction for VLMs, however, they only evict vision KV and preserve all the text KV. TGV-KV evaluates the importance score of each vision KV and is also applicable for purely vision KV eviction. We remove the TPR policy and always keep all the text KV in each layer, in the same setting as these works. The retention budget is defined as the proportion of retained vision KV takes up in the full vision KV. In Table 8, we show the performance comparison. Notably, TGV-KV surpasses the comparison method across all the models and all the budget settings, proving its efficacy in vision KV eviction and further strengthening the design of TVB and TPR.

## C. Extended Visualizations

### C.1. Attention Gap and Dominant Tokens

In Fig. 7, we provide visualizations of the attention map across different VLM layers. The text-text part shows distinct vertical lines, corresponding to the dominant text tokens. Crucially, the text-vision part has low values in all the layers,

empirically validating our observation of the modality gap.

## C.2. Retained KV

We provide more examples of retained KV in Fig. 8. TGV-KV effectively preserves the image patches most related to the question and most critical text counterparts, ensuring the retained KV directly serves the text instruction. This visualization confirms that our method aligns KV cache eviction with the semantic intent of text prompt.

## D. Discussions

### D.1. Towards Long Text and Short Vision.

Under circumstances with long text, all the visual KV may be evicted due to our TPR policy. To evaluate the influence of evicting all vision tokens, we evaluate TGV-KV on ScienceQA (Lu et al., 2022), a VQA dataset with long text. As shown in Fig. 6, the performance drop across all the lengths under small budget are similar, indicating the eviction of evicting vision KV is relative small. Besides, relative studies point out that visual information fuse into text features gradually in the decoder (Neo et al., 2025), therefore TGV-KV still maintains performance for long text and short vision sequences.

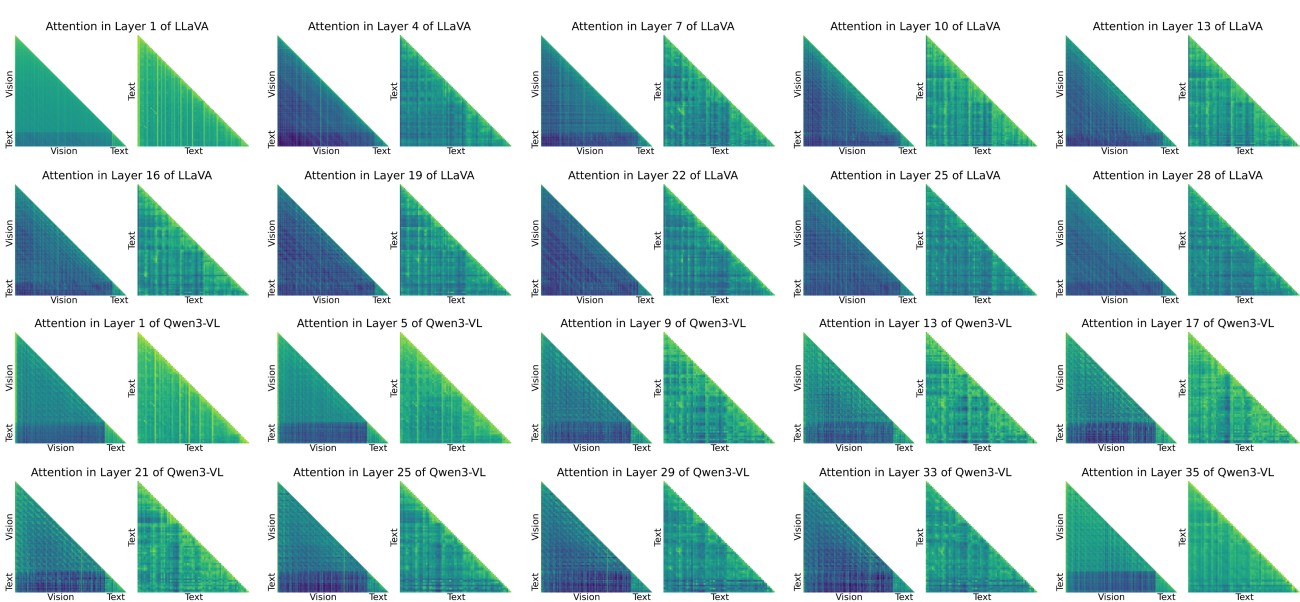

*Figure 7.* **Visualization of Attention Maps.** The image is displayed in log scale.

| Layer 1 | Layer 2 | Layer 6 | Layer 16 | Layer 21 | Layer 32 |
|---|---|---|---|---|---|

USER: Is there a person in the image? ASSISTANT:

USER: Is there a fork in the image? ASSISTANT:

USER: Is there a bird in the image? ASSISTANT:

USER: Is there a sink in the image? ASSISTANT:

*Figure 8.* **Visualization of Retained KV.** The patches most related to the text instruction are circled in violet. Evicted KVs are masked or labelled in gray.

