# OpenReview forum: "TGV-KV: Text-Grounded KV Eviction for Vision-Language Models"
_ICML.cc/2026/Conference — ICML 2026 regular_

### Official Review · Reviewer_d5Za · 2026-03-09

**Soundness:** 2
**Presentation:** 3
**Significance:** 2
**Originality:** 2
**Overall Recommendation:** 4
**Confidence:** 4

**Summary:**

This paper addresses the memory efficiency challenges of Vision-Language Models (VLMs) during long-context inference by introducing TGV-KV (Text-Grounded KV Eviction). The authors identify a "modality gap" where standard language-model KV eviction methods fail because vision tokens and text tokens exhibit different attention patterns. TGV-KV proposes a two-stage mechanism: first, it filters vision tokens based on their attention weights from text tokens (Text-Grounded Filtering), and second, it employs a layer-wise eviction strategy that balances the preservation of text and vision information. Experimental results on benchmarks like DocVQA and ChartVQA demonstrate that TGV-KV significantly reduces KV cache memory (up to 5x) with minimal performance degradation compared to state-of-the-art methods.

**Compliance With Llm Reviewing Policy:**

Affirmed.

**Key Questions For Authors:**

Computational Latency: Could you provide more detailed metrics on the additional latency introduced during the prefill stage by the Text-Weighted Ranking (TWR) and Text-Vision Budgeting (TVB) computations? Specifically, how does this overhead compare to the decoding speedup?

Head-wise Analysis: You noted that head-wise analyses were excluded to maintain parallelism. Have you conducted any preliminary tests to see if head-wise eviction offers significantly better compression, and do you foresee a way to implement it efficiently?

Long-Video Context: While the benchmarks include VQA tasks, how does TGV-KV perform on extremely long-form video understanding tasks where the visual token count is even higher and temporal redundancy is more prevalent?

Generalization: Does the Text-Prioritised Retention (TPR) policy require manual tuning of the budget for different models, or is the mutual information interaction in TVB sufficient to automate this across all architectures?

**Limitations:**

yes

**Strengths And Weaknesses:**

Strengths:
Soundness: The methodology is technically sound, addressing the specific redundancy of visual tokens in VLMs through a grounded approach rather than treating them like pure text tokens. The claims are well-supported by experiments across multiple VLM architectures and diverse benchmarks.

Presentation: The paper is well-structured and clearly explains the motivation behind the "text-grounded" insight. The visualizations of attention patterns effectively illustrate the modality gap.

Significance: Efficient inference is a critical bottleneck for deploying VLMs. Providing a method that maintains performance under high compression ratios (e.g., 20% budget) is highly relevant to both researchers and practitioners.

Originality: The novelty lies in the specific focus on the text-to-vision attention relationship to guide eviction, which differentiates it from general-purpose KV eviction methods like Heavy-Hitter Oracle or StreamingLLM.

Weaknesses:
Evaluation Scope: While the paper tests on several VLM benchmarks, more analysis on extremely long-form video understanding tasks could further demonstrate the robustness of the text-grounding mechanism.

Computational Overhead: The paper mentions memory savings but could provide more detailed metrics on the additional latency introduced by the filtering and ranking process itself during the pre-filling stage.

---

> ### Author Rebuttal · Authors · 2026-03-30
>
> We sincerely appreciate your constructive feedback and for acknowledging the soundness of our text-grounded insight and its strong relevance to efficient VLM deployment.
>
> >> **R-Q1: Computational Latency**
>
> We have provided a detailed prefill latency breakdown under different context lengths with a generation length of 5k. Despite the additional latency, the prefill happens **only once** during generation, and **the overhead of TGV-KV is greatly mitigated by its decoding speedup**. Please refer to R-Q2&W3 for Reviewer `gVfo` for more comparisons with other baselines.
>
> | **Context Length** | KV Budget | Forward (ms) | TVB (ms) | TWR (ms) | TPR (ms) | Others (ms) | Total Prefill (ms) | Throughput Speedup |
> |---|---|---|---|---|---|---|---|---
> | **2k** | 10% | 106.40 | 20.29 | 0.98 | 40.08 | 5.69 | 173.45 | **+6.5%**
> |  | 5% | 106.15 | 14.47 | 0.75 | 11.50 | 7.99 | 140.86 | **+6.7%**
> | **4k** | 10% | 199.85 | 54.33 | 1.89 | 10.38 | 12.56 | 279.01 | **+11.3%**
> |  | 5% | 190.83 | 54.30 | 1.82 | 10.23 | 12.55 | 269.72 | **+12.6%**
> | **8k** | 10% | 426.59 | 246.96 | 6.77 | 74.54 | 32.56 | 787.41 | **+25.4%**
> |  | 5% | 425.86 | 245.42 | 6.37 | 27.78 | 31.55 | 736.98 | **+27.9%**
>
> >> **R-Q2: Head-wise Analysis**
>
> We highly appreciate this insightful suggestion. Accordingly, we implemented a head-wise version of TGV-KV, where TVB assigns varying budgets to different heads, and eviction is performed independently per head. We have the following findings:
>
> - **Slightly improved accuracy:** Most metrics gain a slight improvement, for example, TextCaps grows from 32.6 to 37.2 with 5% budget, indicating that fine-grained head-level budgeting can better capture key information.
> - **Harshly decreased efficiency:** Despite the accuracy improvement, the latency and throughput decrease considerably. This is because head-wise eviction results in varying KV lengths across different heads, which breaks the tensor parallelism of standard GPU matrix multiplications.
>
> To achieve a better accuracy-efficiency trade-off, we currently choose not to take a head-wise alternation. This efficiency issue can be addressed by customized CUDA kernels, which we leave for our future work [1].
>
> **Results for Head-wise**
>
> | Model | Budget | Head-wise | ChartQA | COCO-Caption| DocVQA | TextCaps| TextVQA | VizWiz |
> |---|---|---|---|---|---|---|---|---
> | **LLaVA** | 5% |  | 13.8 | 49.1 | **14.3** | 32.6 | 33.9 | 31.2
> |  | 5% | ✓ | **14.0** | **49.5** | **14.3** | **37.2** | **34.1** | **37.1**
> |  | 20% |  | **18.0** | **55.0** | **21.8** | 87.8 | **47.4** | **54.0**
> |  | 20% | ✓ | 17.7 | **55.0** | 21.1 | **90.4** | 46.8 | 53.8
> | **Qwen3-VL** | 5% |  | **73.1** | **34.4** | 88.0 | 22.5 | 72.0 | **64.4**
> |  | 5% | ✓ | 71.2 | 34.3 | **88.1** | **25.4** | **80.0** | 60.0
> |  | 20% |  | **84.4** | 41.6 | 94.7 | 34.9 | **81.8** | 69.2
> |  | 20% | ✓ | 80.0 | **43.8** | **95.0** | **40.8** | 80.8 | **72.0**
>
> **Latency Comparison**
>
> | Model | Head-wise | Prefill Latency (ms) | Decode Latency (ms) | Throughput (tokens/s) | Total Speedup
> | --- | --- | --- | --- | --- | ---
> | **LLaVA** |  | **787.41** | **19.63** | **45.06** | **1.25**
> |  | ✓ | 1638.57 | 23.45 | 38.46 | 1.07
>
> >> **R-Q3: Long-Video Context**
>
> We have evaluated TGV-KV on MovieChat [2], a benchmark specifically designed for extremely long video understanding and contains videos with more than 12k frames. Remarkably, even **under an extreme budget of 5%, TGV-KV maintains 93.5% accuracy** and a highly considerable score evaluated by LLM-as-a-judge. We will include these long-video evaluation into our updated manuscript.
>
> | Task | 100% | 50% | 20% | 10% | 5%
> | --- | --- | --- | --- | --- | ---
> | MovieChat-Global $^\text{Acc.}$ | 46.1 | 45.3 | 45.2 | 43.4 | 43.1
> | MovieChat-Global $^\text{Score}$ | 2.32 | 2.28 | 2.27 | 2.20 | 2.16
>
> >> **R-Q4: Generalization**
>
> Thanks for highlighting this critic aspect of deployment. We confirm that TGV-KV is fully automated and **requires no manual tuning or heuristic hyperparameters** across different architectures and the mutual information interaction within TVB is entirely sufficient. Our reasons are as follows:
>
> - Automated TVB budgeting: The budget for each layer is assigned by our TVB module, which computes the layer-wise budget $b_l$ on-the-fly based solely on text-vision attention.
> - Deterministic TPR retention: TPR conducts eviction based on a deterministic policy illustrated in Figure 3, which only relies on the budget $b_l$ given by TVB and does not need any hard-coded ratios.
>
> We will highlight this parameter-free and automated nature more explicit in our abstract and method in our revised manuscript.
>
> ---
>
> We sincerely thank the reviewer for the time and constructive feedback. We hope our response has addressed your concerns.
>
> [1] Feng et al., "Ada-KV: Optimizing KV Cache Eviction by Adaptive Budget Allocation for Efficient LLM Inference" (NIPS 2025).
>
> [2] Song et al., "MovieChat: From Dense Token to Sparse Memory in Long Video Understanding" (CVPR 2024).

---

> > ### Author Rebuttal · Reviewer_d5Za · 2026-04-08
> >
> > My concerns have been adequately addressed. I keep the score.

---

### Official Review · Reviewer_4CGT · 2026-03-12

**Soundness:** 3
**Presentation:** 3
**Significance:** 4
**Originality:** 4
**Overall Recommendation:** 5
**Confidence:** 4

**Summary:**

This paper addresses the memory efficiency of Vision-Language Models (VLMs) by proposing TGV-KV, a training-free Key-Value (KV) cache eviction framework specifically tailored for multimodal contexts. The authors identify that standard language model eviction policies fail in VLMs due to the "modality gap", where visual tokens exhibit high redundancy and different importance profiles compared to text. TGV-KV utilizes textual guidance to ground the importance of visual information through three submodules: Text-Vision Budgeting (TVB) for layer-wise allocation, Text-Weighted Ranking (TWR) for cross-modality priority assessment, and Text-Prioritized Retention (TPR) to safeguard critical information. Evaluated across five architectures, the method achieves up to a 95% reduction in cache memory with minimal accuracy loss (retaining 99.2% on VizWiz-VQA) and significantly boosts decoding throughput.

**Compliance With Llm Reviewing Policy:**

Affirmed.

**Final Justification:**

The rebuttal addressed my initial concerns, and I'm maintaining my Accept recommendation. The evaluations comparing AirCache on text-dominant and video tasks demonstrated TGV-KV's effectiveness even in vision-only settings. Additionally, the breakdown of the overhead clarified the one-time TVB prefill latency is worthwhile, given the acceleration in decoding. The explanation of how the architecture generalizes to cross-attention was also sound. Overall, this is a technically solid and practical contribution.

**Key Questions For Authors:**

1. Given that AirCache is a specialized VLM eviction method, could you move its comparative results to the main body to better contextualize TGV-KV’s performance against contemporary multimodal competitors?
2. Prompt Sensitivity: How does the TWR module perform in scenarios with extremely short or non-descriptive prompts (e.g., "Describe the image.") where "dominant text tokens" may be less informative?
3. Computational Overhead: What is the specific inference-time latency added by the TVB budgeting and TWR ranking steps compared to simpler, non-grounded eviction methods?
4. Architecture Generalization: Do you anticipate that the textual grounding principle would require significant modification for models that use cross-attention adapters rather than a shared transformer backbone?

**Limitations:**

The authors provide an honest evaluation of the method's performance across different architectures. However, a more explicit discussion of the computational overhead introduced by the real-time mutual information and weighting calculations would be beneficial.

**Strengths And Weaknesses:**

Strengths

Soundness: The paper provides a systematic analysis of the multimodal attention pattern, identifying three key differences—modality redundancy, diverse importance, and textual guidance—that lead to the failure of standard LLM eviction methods in VLMs.

Presentation: The overall narrative is easy to follow. The "Key Insights" section (3.1–3.3) provides a strong logical bridge from initial observations to the final method design.

Significance: The framework offers substantial practical utility, reducing KV cache memory from 3.91 GB to 0.20 GB for an 8k context, which is vital for deploying long-context VLMs on resource-constrained devices.

Originality: The "text-grounded" perspective is a novel contribution to the field of multimodal compression. The TWR module's approach of using "dominant text tokens" to weight vision importance effectively bridges the modality gap.

Weaknesses

Soundness: While multiple baselines are evaluated, the comparison to AirCache, a concurrent and highly relevant VLM-specific method, is relegated to the appendix. This placement makes it difficult to fully assess TGV-KV's state-of-the-art standing in the main text.

Presentation: Certain mathematical formulations in section 3.3, specifically Equations 7-10 regarding layer-wise budget allocation, are quite dense. Clarifying variable definitions in these equations would improve reproducibility.

Significance: The evaluation mainly focuses on decoder-only VLMs like LLaVA and Qwen. The paper would be strengthened by discussing or testing applicability to other architectures, such as encoder-decoder models or those utilizing different adapter mechanisms.

---

> ### Author Rebuttal · Authors · 2026-03-30
>
> We thank the reviewer for the positive assessment, particularly for recognizing our logical presentation, the novelty of our text-grounded perspective, and the significant practical utility for long-context VLMs.
>
> >> **R-Q1: Results on AirCache**
>
> We initially put the results of AirCache in the appendix rather the main body mainly for the following reasons:
>
> - AirCache is a **vision-only** eviction methods (preserve all text KV), while TGV-KV and comparisons in Table 2&3 are methods for unified eviction (evict both vision and text KV).
> - They use fundamentally different budget calculation settings, and direct comparison in the main text required adapting TGV-KV to a vision-only setting (detailed in App. B.4).
>
> To address your point, we provide extended evaluations on text-dominant tasks and video tasks. TGV-KV also keeps its outstanding performance under vision-only settings, proving the effectiveness of TWB in assessing vision importance.
>
> **Results for Text-Dominant Tasks**
>
> | Budget | Method | TextCaps $^\text{CIDEr}$ | TextCaps $^\text{ROUGE-L}$ | COCO-Cap $^\text{CIDEr}$ | COCO-Cap $^\text{ROUGE-L}$ |
> | --- | --- | --- | --- | --- | --- |
> | 50% | AirCache | 81.2 | 44.5 | 104.2 | 54.7 |
> |  | TGV-KV | **100.1** | **47.1** | **109.2** | **55.2** |
> | 20% | AirCache | 57.4 | 41.5 | 93.5 | 52.6 |
> |  | TGV-KV | **93.2** | **46.3** | **109.9** | **55.3** |
> | 10% | AirCache | 45.6 | 39.8 | 85.8 | 51.2 |
> |  | TGV-KV | **80.1** | **44.3** | **106.6** | **54.7** |
>
> **Results for Video-TT**
>
> | Method | Para. 50% | Wrong. 50% | Para. 20% | Wrong. 20% | Para. 10% | Wrong. 10% |
> | --- | --- | --- | --- | --- | --- | --- |
> | AirCache | **21.4** | 26.9 | **20.4** | 26.8 | 18.7 | 25.8 |
> | TGV-KV | 20.8 | **27.1** | **20.4** | **27.2** | **20.7** | **25.9** |
>
> >> **R-Q2: Prompt Sensitivity**
>
> TWR assigns weight based on text significance and we show that it is still robust under short or non-descriptive prompts for the following reasons:
>
> - The prompt of **text-dominant tasks (TextCaps, COCO-Caption) in our evaluations are almost all non-descriptive ones** that require the model to write a caption, and the results in Table 3&8 show the superiority of TGV-KV.
> - Results on **different prompt lengths** (R-Q1 for Reviewer `gVfo` ) also prove TWR module’s effectiveness when handling short and long prompts.
> - We conduct a qualitative visualization for extremely short prompts in the table below. It is interesting that in middle layers with very low budget, the text tokens of the prompt are uniformly preserved across layers, while in Figure 5 with a descriptive prompt, the ‘’tennis ball’’ is kept across all the layers. These studies show that TWR maintains high universality and diversity.
>
>
>     | Layer | Prompt |
>     | --- | --- |
>     | 1 | `USER: Describe this image. ASSISTANT:` |
>     | 6 | `USER:` Des`cribe this` image. ASSISTANT`:` |
>     | 10 | `USER: Des`cribe `this image`. A`SS`IST`ANT:` |
>     | 16 | `USER: Describe this image. A`SSISTANT`:` |
>     | 32 | `USER: Describe this image. ASSISTANT:` |
>
> >> **R-Q3: Computational Overhead**
>
> Thank you for requesting this detailed breakdown. We have measured the latency of each component during the prefill stage and presented the results in R-Q1 for Reviewer `d5Za`. In summary:
>
> - **TWR** is highly lightweight which adds < 1% of total prefill latency even at 8k context, as it is highly parallel matrix operation on GPU.
> - **TVB** is the main source of overhead because it requires aggregating multi-modal matrices from all the layers. However, as shown in the “Throughput Speedup” column, this one-time overhead is fully mitigated by the accelerated decoding, **bringing up to 28% speedup**, which is faster than the compared baselines in R-Q2&W3 to Reviewer `gVfo`.
>
> We will include this transparent discussion of computational overhead in the revised manuscript.
>
> >> **R-Q4: Architecture Generalization**
>
> When transferring to VLMs using cross-attention adaptors (e.g., Flamingo), thanks to their more explicit cross-modal interaction paradigm, TGV-KV can be integrated by the following minor modifications:
>
> - TVB: In cross-attention layers where text are query and vision are key, the text-vision attention is the whole attention matrix $A^{\text{cross}} \in \mathbf{R}^{N_t \times N_v}$, the sum of text-vision attention is a fixed value of $N_t$ due to the softmax. Therefore, we turn to directly **use the gating score** $b_l \propto |tanh(\alpha_l)|$ since the learned gating scalar $|tanh(\alpha_l)|$ controls the injection magnitude of vision feature. This adoption is even more computation-friendly since it only contains several scalar operations.
> - TWR: No modification required since the column sum of cross attention is not affected by the aforementioned issue.
>
> ---
>
> We deeply appreciate your insightful comments, which have significantly strengthened our paper. Thank you for your time and effort.
>
> [1] Alayrac et al., "Flamingo: a Visual Language Model for Few-Shot Learning" (NIPS 22).

---

> > ### Author Rebuttal · Reviewer_4CGT · 2026-04-02
> >
> > AirCache Comparison: The extended evaluations on text-dominant and video tasks clearly demonstrate TGV-KV's effectiveness even in vision-only settings. Providing this data significantly strengthens the baseline comparisons and justifies your architectural choices.
> >
> > Prompt Sensitivity: The clarification regarding the non-descriptive nature of the TextCaps and COCO-Caption prompts effectively answers my concern. The qualitative visualization showing preservation of text tokens across layers for extremely short prompts is an interesting finding and confirms the robustness of the TWR module.
> >
> > Computational Overhead: The breakdown of prefill latency is exactly what was needed. The tradeoff, where the one-time TVB overhead is entirely offset by the accelerated decoding, is a strong practical selling point. I am glad to hear this discussion will be included in the revised manuscript.
> >
> > Architecture Generalization: Your explanation of how TVB can be adapted for cross-attention mechanisms (e.g., Flamingo) by utilizing the learned gating score $b_l \propto |\tanh(\alpha_l)|$ is mathematically sound. It is encouraging that this adaptation is even more computationally friendly.
> >
> > Overall, the rebuttal reinforces the strength and originality of the TGV-KV framework. I will be maintaining my rating of 5: Accept.

---

### Official Review · Reviewer_gVfo · 2026-03-12

**Soundness:** 3
**Presentation:** 3
**Significance:** 2
**Originality:** 2
**Overall Recommendation:** 4
**Confidence:** 4

**Summary:**

This paper studies KV cache eviction for vision-language models. The core claim is that VLM eviction should be designed around text-grounded multimodal interactions rather than copied from LLM-only heuristics. The proposed method, TGV-KV, combines text-vision budgeting, text-weighted ranking of visual tokens, and text-prioritized retention. The paper reports strong efficiency gains with relatively small quality loss across image, video, and text-heavy settings. This work sits in the recent VLM efficiency literature on KV compression and token reduction [ref-1, ref-2].

> [ref-1] Li et al., "SnapKV: LLM Knows What You are Looking for Before Generation" (2024).
>
> [ref-2] Chen et al., "An Image is Worth 1/2 Tokens After Layer 2" (ECCV 2024).

**Compliance With Llm Reviewing Policy:**

Affirmed.

**Final Justification:**

The matched-hardware comparison against SnapKV and H2O (Q2/W3) was the most useful part of the rebuttal, and Q3 and Q4 were handled cleanly. W1 stays at the correlation level: the TWR ranking ablation shows rankings matter more than raw attention scores, but says nothing about whether tokens with persistently low early-step attention are still needed later in multi-step reasoning and get evicted before they matter. I'm holding at 4.

**Key Questions For Authors:**

Q1. How sensitive are the results to prompt style and prompt length, especially on text-heavy tasks?

Q2. Can you provide matched latency, memory, and throughput comparisons against the strongest recent VLM-specific KV compression baselines on the same hardware?

Q3. Does TPR over-protect text in OCR or document settings where some visual tokens may be more important than parts of the prompt?

Q4. Appendix B.4 / Table 9 appears to use the AirCache-style setting, with TPR removed, all text KV retained, and the budget applied only to vision KV. Could the authors clarify this regime explicitly in the comparison?

**Limitations:**

The main limitation is that the efficiency story is much stronger than the explanatory story. The paper gives good evidence that the method works, but weaker evidence that the modality-gap analysis fully explains why it works. The current experiments are also not enough to support the broader claim that the same budgeting logic will transfer cleanly to newer native multimodal models.

**Strengths And Weaknesses:**

## Strengths

The paper is motivated by a practical and timely problem, the method is easy to follow, and the empirical story is strongest on efficiency. Overall, the work gives a fairly coherent case that the proposed design is useful in practice and not just a narrow benchmark improvement.

## Weaknesses

- **[W1] Causal support**: The modality-gap analysis is interesting, but Figure 1 and Table 1 still support a correlation-driven story more than a causal one.
- **[W2] Scope of claims**: The universality claim near the end is too strong for the current model coverage. The experiments are broad, but still limited to a fairly specific slice of VLM architectures.
- **[W3] Missing matched baseline comparison**: Table 4 compares TGV-KV against vanilla decoding, not against the strongest competing eviction methods under matched hardware and serving settings. That makes it hard to tell how much of the systems gain is really state-of-the-art relative, rather than simply better than no compression [ref-1, ref-2].
- **[W4] Writing quality**: Some claims are stated more strongly than the evidence supports, especially when the paper moves from broad empirical trends to universal conclusions.

> [ref-1] Li et al., "SnapKV: LLM Knows What You are Looking for Before Generation" (NeurIPS 2024).
>
> [ref-2] Chen et al., "An Image is Worth 1/2 Tokens After Layer 2" (ECCV 2024).

---

> ### Author Rebuttal · Authors · 2026-03-30
>
> We greatly appreciate your constructive review and for highlighting that our work addresses a timely problem with strong efficiency gains and a coherent design.
>
> >> **R-Q1: Sensitivity to Prompt Styles and Lengths**
>
> Thanks for your insightful question. We conduct a detailed quantitative analysis on the ScienceQA and categorize the validation samples into groups with similar instruction lengths. We have presented preliminary results in Figure 6 and extended results below.
>
> **Results for ScienceQA with Different Prompt Lengths**
>
> | **Budgets** | 0-50 | 50-100 | 100-150 | 150-200 | 200-250 | 250+ |
> | --- | --- | --- | --- | --- | --- | --- |
> | Samples | 886 | 532 | 327 | 46 | 147 | 79 |
> | 100% | 70.5 | 61.5 | 51.4 | 39.1 | 53.7 | 81.0 |
> | 30% | 70.5 | 61.5 | 51.4 | 39.1 | 53.7 | 79.8 |
> | 20% | 70.5 | 61.5 | 51.4 | 39.1 | 53.7 | 79.8 |
> | 10% | 69.9 | 59.9 | 50.8 | 38.5 | 53.1 | 78.0 |
>
> Our conclusions are that:
>
> - Under moderate budgets, TGV-KV achieves nearly zero degradation across all length groups.
> - Even under a highly restricted 10% budget, the performance drop remains marginal, less than 2% across all the prompt lengths.
>
> Please refer to R-Q2 for Reviewer `4CGT`  for a qualitative example of non-descriptive style prompt. These results show that TGV-KV is robust to different style and length, even on multi-modal datasets with heavy text.
>
> >> **R-Q2&W3: Balanced Efficiency Comparisons**
>
> We have conducted a comprehensive end-to-end evaluation against the mentioned token pruning and KV cache eviction methods in the following table. The results are obtained on Pro 6000. We have two key findings:
>
> - **Similar system acceleration:** All the KV cache methods share a similar throughput acceleration under same budget. Although TGV-KV incurs a higher prefill latency, its decoding speedup completely compensate this issue.
> - **Distinct model performance:** Under an extreme budget, most comparisons show catastrophic collapse, e.g., SnapKV and H2O maintain only 1.2 and 2.1 on DocVQA with 5% KV, while TGV-KV preserves robust accuracy.
>
> | Budget | Method | Prefill Latency (ms) | Decode Latency (ms) | Throughput (tokens/s) | Total Speedup | DocVQA |
> | --- | --- | --- | --- | --- | --- | --- |
> | 100% | Vanilla-Eager | 444.51 | 25.23 | 35.93 | 1.00 | 23.9 |
> |  | Vanilla-FA2 | 235.30 | 29.48 | 36.69 | 1.02 | 23.9 |
> | 10% | FastV* | 441.43 | 25.21 | 36.10 | 1.00 | - |
> |  | SnapKV | 462.21 | 20.36 | 44.87 | 1.25 | 5.1 |
> |  | H2O | 445.35 | 19.85 | 45.07 | 1.25 | 3.9 |
> |  | TGV-KV | 787.41 | 19.63 | 45.06 | 1.25 | 19.2 |
> | 5% | FastV* | 441.12 | 25.21 | 36.10 | 1.00 | - |
> |  | SnapKV | 448.46 | 20.42 | 45.56 | 1.27 | 1.2 |
> |  | H2O | 444.58 | 20.35 | 45.49 | 1.27 | 2.1 |
> |  | TGV-KV | 736.98 | 20.32 | 45.94 | 1.27 | 14.3 |
>
> `*` FastV is a method that only prunes vision tokens, and the budget for FastV stands for the retained vision tokens.
>
> >> **R-Q3: OCR Task Evaluations**
>
> Related studies have proposed that the vision information flows into the text during the shallow layers [1], while a small set of key tokens (so-called "register tokens") hold the key representations of the image [2]. These studies indicate that the model is likely to generate correct answers even when most correlated vision tokens or KV are removed. We have conducted evaluations under lower budget on OCRBench. TGV-KV demonstrates superior OCR ability against other methods.
>
> **Results for OCRBench**
>
> | Budget | TGV-KV | StreamingLLM | PrefixKV | PyramidKV | SnapKV | H2O |
> | --- | --- | --- | --- | --- | --- | --- |
> | 10% | **16.2** | 7.7 | 0.5 | 1.0 | 0.7 | 0.7 |
> | 20% | **19.2** | 10.4 | 1.4 | 2.8 | 0.8 | 0.7 |
>
> >> **R-Q4: Explicit Explanation of AirCache Settings**
>
> AirCache is an eviction method for vision KV, it **only evicts vision KV while preserving all the text KV**. TGV-KV is a method for both vision and text KV and **evicts text as well as vision**. For a fair comparison, we also preserve all the text KV and evict vision KV (we denote this as "with TPR removed" since all the text KV are preserved manually) during the evaluation in Table 9. For example, in an input sequence consisting of 500 vision tokens and 200 text tokens, the retention of 5% in this setting yields an equivalent sequence of 25 vision tokens and 200 text tokens ("the budget applied only to vision KV").
>
> >> **R-W1: Causal support**
>
> We have conducted a controlled ablation study on TWR and attention score, please refer to our R-Q1 for Reviewer `1HCb`.
>
> >> **R-W2&W4:  Scope of claims and Writing quality**
>
> We thank the reviewer for pointing out this limitation. Our method is evaluated on popular AR VLMs and its “universality” should be bounded within this architecture paradigm. We will thoroughly tone down our language.
>
> ---
>
> We sincerely thank the reviewer for the valuable feedback that helped improve our work.
>
> [1] Zhang et al., “Cross-modal Information Flow in Multimodal Large Language Models” (CVPR 2025).
>
> [2] Darcet et al., “VISION TRANSFORMERS NEED REGISTERS” (ICLR 24).

---

> > ### Author Rebuttal · Reviewer_gVfo · 2026-04-04
> >
> > The matched-hardware comparison against SnapKV and H2O was the most valuable addition in the rebuttal, and Q3, Q4 were handled well.
> >
> > However, reading Reviewer 1HCb's discussion raised a new concern: the causal ablation shows that TWR rankings matter more than raw attention scores, which is a useful data point, but it doesn't quite address whether attention-based signals can capture tokens that only become important later in multi-step generation. A token with consistently low early-step attention could still be critical for long-range reasoning and would be evicted regardless. This feels like a meaningful gap, especially at extreme budgets where the method is most brittle.
> >
> > Combined with W1 from my original review still sitting at the correlation level, this is why I'm adjusting my score. I'd like to ask: can the authors speak more directly to the delayed-utility question, perhaps with a generation-step-level analysis of which tokens get evicted and when?

---

> > > ### Author Response · Authors · 2026-04-05
> > >
> > > We sincerely thank you for engaging so deeply with our work. We have conducted generation-step-level analyses on the evicted tokens and show that TGV-KV does not suffer from this delayed-utility question quantitatively and qualitatively.
> > >
> > > >> **Hit Rate Analysis Prove High Coverage**
> > >
> > > To trace whether tokens evicted by TGV-KV suddenly become important in later decoding steps, we conducted a step-by-step analysis on tokens recalled at different decoding steps. Specifically, we measured the hit rate, the percentage of the actual Top-K highest-attention tokens required at decoding step $t$ (identified by a full-KV Oracle baseline) that were successfully retained by TGV-KV at the prefill stage. The results show that **the overlap between TGV-KV's prefill selection and the actual tokens needed during subsequent late decoding steps remains highly stable and consistently high across the entire generation sequence**, and the phenomenon where a "forgotten" visual token suddenly becomes critical is empirically very rare. We credit this consistency and stability to the following two reasons.
> > >
> > > | Budget | Top-K | Prompt Length | Step 1 | Step 10 | Step 20 | Step 30 | Step 50 | Step 75 | Step 100 |
> > > | :-: | :-: | --- | :-: | :-: | :-: | :-: | :-: | :-: | :-: |
> > > | **50%** | **20%** | **0-100** | 100.0 | 100.0 | 99.1 | 98.1 | 97.2 | 99.0 | 98.2 |
> > > |  |  | **100-200** | 100.0 | 99.4 | 98.3 | 97.9 | 98.3 | 96.0 | 95.1 |
> > > |  |  | **200-300** | 100.0 | 99.6 | 99.7 | 98.3 | 98.4 | 96.5 | 95.4 |
> > > |  |  | **300+** | 100.0 | 100.0 | 99.6 | 98.2 | 97.1 | 96.5 | 94.6 |
> > > |  | **10%** | **0-100** | 100.0 | 100.0 | 98.7 | 99.4 | 96.1 | 92.5 | 82.5 |
> > > |  |  | **100-200** | 100.0 | 98.6 | 99.0 | 95.4 | 96.9 | 88.4 | 84.8 |
> > > |  |  | **200-300** | 100.0 | 98.9 | 100.0 | 96.2 | 94.6 | 92.5 | 87.3 |
> > > |  |  | **300+** | 100.0 | 100.0 | 99.1 | 98.2 | 93.4 | 91.2 | 83.5 |
> > > | **20%** | **20%** | **0-100** | 100.0 | 100.0 | 98.3 | 98.3 | 96.8 | 98.3 | 97.3 |
> > > |  |  | **100-200** | 100.0 | 98.0 | 96.8 | 97.6 | 97.3 | 97.4 | 94.8 |
> > > |  |  | **200-300** | 100.0 | 99.1 | 98.9 | 98.8 | 96.3 | 95.5 | 95.0 |
> > > |  |  | **300+** | 100.0 | 98.6 | 98.8 | 97.8 | 96.2 | 98.7 | 94.3 |
> > > |  | **10%** | **0-100** | 100.0 | 100.0 | 98.4 | 98.8 | 95.7 | 85.7 | 80.0 |
> > > |  |  | **100-200** | 100.0 | 96.9 | 93.8 | 94.0 | 93.4 | 86.3 | 80.6 |
> > > |  |  | **200-300** | 100.0 | 97.5 | 96.8 | 95.9 | 90.8 | 87.6 | 85.1 |
> > > |  |  | **300+** | 100.0 | 96.6 | 98.6 | 96.0 | 91.1 | 91.6 | 80.0 |
> > >
> > > > **1. Prefill Attention Captures Delayed Utility**
> > >
> > > As revealed by related attention-driven works, the model forms a strong internal representation of the final answer during the prefill stage [1,2]. In VLMs, no matter whether the text prompt is a descriptive one (Fig. 5 in the main paper) or a non-descriptive one (R-Q2 for Reviewer `4CGT`), our TWR module **precisely identifies the highly information-centric and instruction-related text and vision KVs, which are highly likely to be critical for future decoding**. Therefore, a combination of prefill attention and our TWR module effectively captures the delayed utility.
> > >
> > >
> > > > **2. TPR Preserves Potential Important KVs at Early Stages**
> > >
> > > Long-range reasoning like CoT primarily **relies on the text reasoning trajectory** in previous steps, rather than repeatedly revisiting uninformative visual background patches [3]. This also explains why conventional LLM-oriented methods fail in VLM, as they indiscriminately evict text tokens that will be recalled in future decoding steps. Empirically, text tokens are more important than vision tokens in VLM [4], even if **some of the text tokens may demonstrate lower attention score than several salient vision tokens at the early decoding stage**. Rather than evicting text tokens based solely on early attention, our TPR design **preserves the text tokens for future access**, avoiding the risk of late-stage reasoning collapse.
> > >
> > > ---
> > > We deeply appreciate your rigorous review process. Tracing the step-by-step utility significantly strengthens our claims, and we hope this explicit evidence fully addresses your new concern and restores your confidence in supporting our work.
> > >
> > > [1] Li et al., SnapKV: LLM Knows What You are Looking for  Before Generation, NIPS 2024.
> > >
> > > [2] Zhang et al., H2O: Heavy-Hitter Oracle for Efficient Generative Inference of Large Language Models, NIPS 2023.
> > >
> > > [3] Zhang et al., Cross-modal Information Flow in Multimodal Large Language Models, CVPR 2025.
> > >
> > > [4] Chen ti al., An Image is Worth 1/2 Tokens After Layer 2: Plug-and-PLay Acceleration for VLLM Inference, ECCV 24.

---

### Official Review · Reviewer_1HCb · 2026-03-19

**Soundness:** 3
**Presentation:** 3
**Significance:** 3
**Originality:** 3
**Overall Recommendation:** 4
**Confidence:** 4

**Summary:**

This paper tackles inference-time KV cache eviction for vision-language models (VLMs), motivated by two practical issues: (i) KV cache grows linearly with context length, becoming a dominant memory bottleneck during long generation; and (ii) VLM inputs often contain a large number of redundant visual tokens, making naive caching inefficient. The authors further argue that LLM-oriented eviction heuristics do not transfer well to VLMs, because cross-modal attention patterns differ substantially from purely textual settings (“modality gap”), so direct adoption can remove modality-critical KV entries and cause sharp accuracy drops at small budgets.
To address this, the paper proposes TGV-KV, a training-free, plug-and-play KV eviction framework that uses text as an anchor signal to decide which KV entries to retain, with a particular focus on preserving cross-modal information flow during decoding. The method has three main components:
	TVB (Text-to-Vision Budgeting): allocate per-layer KV retention budgets based on aggregated text→vision attention, under the intuition that layers with stronger cross-modal interactions require more KV capacity.
	TWR (Text-Weighted Ranking): compute a text-token importance weight (estimated from text→text attention statistics) and use it to reweight text→vision attention, producing a more robust ranking criterion for evicting visual KV.
	TPR (Text-Prioritized Retention): preferentially retain text KV whenever possible, only evicting text KV in extremely constrained budgets, based on the observation that dropping text KV tends to harm generation disproportionately.
The paper evaluates TGV-KV on multiple VLM backbones (e.g., LLaVA variants and Qwen-VL variants) and a range of tasks including image VQA/captioning and long-context/video settings. Results show that TGV-KV generally outperforms prior eviction baselines, especially under aggressive retention ratios (e.g., 5%), and provides tangible efficiency benefits (reduced memory and improved throughput/latency). Overall, the submission positions TGV-KV as a practical inference optimization that better respects modality-specific KV importance by leveraging textual guidance.

**Compliance With Llm Reviewing Policy:**

Affirmed.

**Key Questions For Authors:**

1.	Attention statistics extraction: How are the TV and TT attention summaries computed in practice under FlashAttention/efficient attention kernels where attention matrices are not materialized? What is the additional memory/time overhead during prefill and decode?
2.	Causal validation of importance signals: Do you have diagnostic evidence that TWR’s weighted attention ranking correlates with causal importance (e.g., controlled ablation/deletion of selected vs. non-selected tokens, gradient-based importance agreement, or attention-rollout comparisons)?
3.	Robustness under vision-critical tasks: Under extremely small budgets, does TPR ever lead to near-complete eviction of visual KV? If so, how does performance behave on tasks requiring repeated visual grounding (fine-grained localization, counting, OCR-heavy, multi-image comparison)? Would a minimum per-layer/per-modality quota help?
4.	Baseline fairness and tuning protocol: Were baseline methods tuned per model/task at each retention ratio (e.g., window sizes, heavy-hitter thresholds, layer schedules)? Did all methods use the same “keep first 4 + last 1 token” constraint, and how sensitive are results to that choice?
5.	Closest related method comparison: Can you provide a more prominent, apples-to-apples comparison with the most relevant VLM-specific method (e.g., AirCache) in the main tables and discuss where each method wins/loses?

**Limitations:**

The paper discusses some limitations (e.g., certain cases where text dominates and vision KV may be heavily evicted), but it does not fully characterize potential failure modes in vision-critical settings (OCR/counting/localization, multi-image/video comparisons) nor does it clearly quantify how often the method collapses vision retention under tight budgets. It would also help to discuss how the approach might interact with different VLM architectures (early vs. late fusion) and provide mitigation strategies (e.g., minimum vision quota).

**Strengths And Weaknesses:**

Strengths:
The evaluation spans several model families (multiple LLaVA variants and Qwen-VL variants), multiple tasks (VQA, captioning, video/long-context), and multiple budgets (including very tight retention). This breadth reduces the chance that gains are due to a single cherry-picked configuration.
The paper provides a coherent chain from observations → algorithmic components (TVB/TWR/TPR). In particular, TPR is well motivated by the high sensitivity of generation to text-token eviction.
Since no fine-tuning is required, the method is less likely to introduce confounding factors from training procedures, and it is easier to verify improvements as purely inference-time effects.
The componentization (TVB, TWR, TPR) makes it possible to understand which part contributes, and the paper includes supporting analyses such as modality-gap visualizations.

Weaknesses:
The core scoring signals rely on attention statistics (TV mass, TT column sums). There is a known mismatch between attention and causal contribution; tokens with low attention can still be essential via residual pathways or earlier fusion, and high attention can be superficial. Without causal validation, it is unclear when the heuristic might fail.
Layer budget heuristic may be model- and architecture-dependent. TVB assumes that higher aggregated text→vision attention implies higher need for KV retention in that layer. This may vary with fusion style (early vs. late), model depth, or attention head specialization. The paper would be stronger with evidence that TVB generalizes across architectures beyond the tested set (or at least a discussion of expected failure modes).
Potential mismatch between “importance for next-token” vs. “importance for multi-step reasoning.” KV eviction is applied across generation steps; some tokens might be essential only later in the chain of thought or for long-range reference. Attention-based ranking from a single step (or aggregated in a limited way) may not capture delayed utility.
Extreme-budget behavior not fully characterized. In very low retention regimes, the method can effectively prioritize text to the point where vision KV becomes extremely sparse. The paper argues this is often acceptable, but it remains unclear how often and under what conditions it leads to catastrophic failures on visually grounded prompts.

---

> ### Author Rebuttal · Authors · 2026-03-30
>
> We sincerely thank the reviewer for the thoughtful feedback and for recognizing our extensive evaluation, coherent methodology, and the practical utility of our TPR component. Below we address your concerns in detail.
>
> >> **R-Q1: FlashAttention Issues and System Overhead**
>
> While TGV-KV currently requires materializing attention matrices , **the overall end-to-end acceleration of TGV-KV is substantially superior to using pure FA**. As detailed in the comprehensive latency breakdowns provided to Reviewers `gVfo` and `d5Za`, the results suggest that: **The prefill penalty of incompatibility with FA2 is completely compensated by the decoding acceleration,** which brings a 27.9% throughput acceleration under 8k context with 5% budget compared with FA2.
>
> >> **R-Q2: Causal Validation of Important Signals**
>
> Emperically, the attention-based methods have been widely adopted the KV and token importance. To validate causality, we replace top 10% important KV measured by attention or TWR with random the unselected KV. The results show that when removing TWR KV, the relative performance drops more significantly, proving that the TWR better weighs the real importance of KV than pure attention.
>
> **Controlled ablation of KV**
>
> | Imp. Score | TextCaps $^\text{CIDEr}$ | ChartQA | COCO-Cap $^\text{CIDEr}$ | TextVQA |
> | --- | --- | --- | --- | --- |
> | -10% Attention | -12.3% | -16.3% | -20.6% | -15.4% |
> | -10% TWR | **-34.2%** | **-26.4%** | **-35.9%** | **-25.1%** |
>
> >> **R-Q3: Robustness under Vision-critical Tasks**
>
> Under extremely small budgets, TPR may evict all the visual KV in middle layers. However, since VLMs integrate vision information in text tokens in the shallow layers [1, 2],the model remains robust. To quantify the performance on repeated visual grounding, we extend our evaluations as the reviewer suggests. The results show that TGV-KV achieves nearly lossless performance even at a 10% budget.
>
> **Results for Vision-Critical Tasks**
>
> | Type | Task | 100% | 50% | 10% | 5% |
> | --- | --- | --- | --- | --- | --- |
> | Localization | BLINK-localization | 43.4 | 42.6 | 43.4 | 42.6 |
> | Counting | BLINK-counting | 37.5 | 37.5 | 37.5 | 35.8 |
> | Multi-Image | BLINK-multi img reasoning | 54.1 | 54.1 | 54.1 | 54.1 |
> | OCR | OCRBench | [Refer to | R-Q3 for | Reviewer | `gVfo`] |
>
> Furthermore, we tested your insightful suggestion of maintaining a minimum per-modality quota of 2% KV. As shown below, this strategy improves performance especially under lower budgets. We will include this strategy and a detailed discussion on failure modes in the final version.
>
> **Results for Minimum Per-modality Quota**
>
> | Mini. Quota | Budget | ChartQA | TextVQA | TextCaps $^\text{CIDEr}$ | BLINK-counting |
> | --- | --- | --- | --- | --- | --- |
> |  | 5% | 15.5 | 44.5 | 32.6 | 35.8 |
> | ✓ | 5% | 16.3 | 45.3 | 39.4 | 37.5 |
> |  | 20% | 18.0 | 47.4 | 87.8 | 37.5 |
> | ✓ | 20% | 17.8 | 47.4 | 87.8 | 37.5 |
>
> >> **R-Q4: Baseline Settings**
>
> The baseline methods are **not** tunned per task/model and we use the implementation from [3] with hyperparameter settings identical to each one’s original paper. We manually keep the first and last tokens because most compared methods (SnapKV, H2O, PrefixKV, ElasticCache) specially protects the last tokens since they contain latest information while have less accumulated attention and are vulnerable for attention-based methods. However, as shown below, ablating this heuristic reveals that TGV-KV's performance remains highly robust without it, proving our method does not rely on manual tuning.
>
> **Sensitivity to Keep First 4 + Last 1 Token**
>
> | Preserve 4+1 | Budget | ChartQA | DocVQA | TextVQA | COCO-Cap $^\text{ROUGE-L}$ |
> | --- | --- | --- | --- | --- | --- |
> | ✓ | 10% | **15.5** | **19.2** | 44.5 | **52.6** |
> |  | 10% | 15.4 | 19.1 | **44.7** | **52.6** |
> | ✓ | 20% | **18.0** | **21.8** | **47.4** | **55.0** |
> |  | 20% | **18.0** | **21.8** | 47.3 | **55.0** |
>
> >> **R-Q5: Closest Method Comparison**
>
> We did not place AirCache into the main tables because it utilizes fundamentally different eviction settings which evicts only vision KV (refer to R-Q4 for Reviewer `gVfo`). We have provided more comparisons on text-dominant tasks and video tasks in R-Q1 for Reviewer `4CGT`. It is notable that TGV-KV achieves better results under almost all the settings, and the superiority is more pronounced when the budget is lower. We attribute this to our TVB module, which better captures cross-modal information interactions than AirCache's distribution-skewness metrics.
>
> ---
>
> We sincerely thank the reviewer for the time and constructive feedback. We hope our response has addressed your concerns.
>
> [1] Chen et al., "An Image is Worth 1/2 Tokens After Layer 2: Plug-and-PLay Acceleration for VLLM Inference"(ECCV 2024).
>
> [2] Zhang et al., "Cross-modal Information Flow in Multimodal Large Language Models" (CVPR 2025).
>
> [3] Liu et al., "Efficient Inference of Vision Instruction-Following Models with Elastic Cache" (ECCV 2024).

---

### Decision · Program_Chairs · 2026-04-30

**Decision:**

Accept (regular)

**Comment:**

This paper proposes TGV-KV, a training-free Key-Value (KV) cache eviction framework for Vision-Language Models (VLMs) that leverages text-grounded signals to effectively compress the KV cache while preserving critical cross-modal information. Reviewers agreed that the submission addresses efficiency bottleneck in VLM deployment, praising its evaluation across multiple model families, its empirical efficiency gains, and its component design (TVB, TWR, TPR). Initial reviewer concerns centered around the correlational attention statistics rather than causal importance, the lack of baseline comparisons (e.g., against AirCache and SnapKV), and failure modes on vision-critical or long-video tasks. During the rebuttal phase, the authors provided additional experiments, including matched-hardware latency comparisons, causal ablation studies, and performance metrics on long-video (MovieChat) and OCR tasks, which resolved the majority of the concerns. Although one reviewer noted that the provided step-level analysis does not completely eliminate theoretical concerns regarding the delayed utility of early-evicted tokens, the empirical evidence demonstrates the method's effectiveness, robustness at compression, and efficiency. The paper is recommended for acceptance.